# Aerosol sources in the western Mediterranean during summertime: A model-based approach

Mounir Chrit[1], Karine Sartelet[1], Jean Sciare[2,6], Jorge Pey[3*], José B. Nicolas[4], Nicolas Marchand [3], Evelyn Freney[4], Karine Sellegri[4], Matthias Beekmann[5], and François Dulac[2]

[1] CEREA, joint laboratory Ecole des Ponts ParisTech - EDF R&D, Université Paris-Est, 77455 Champs sur Marne, France.

[2] LSCE, CNRS-CEA-UVSQ,IPSL,Université Paris Saclay, Gif-sur-Yvette, France

[3] Aix Marseille University-CNRS, LCE, Marseille, France

[4] LAMP, UMR CNRS-Université Blaise Pascal, OPGC, Aubière, France

[5] LISA, UMR 7583, Université Paris Diderot-Université Paris-Est Créteil, IPSL, Créteil, France

[6] EEWRC, The Cyprus Institute, Nicosia, Cyprus

[*] Now at the Spanish Geological Survey, IGME, 50006 Zaragoza, Spain

*Correspondence to:* Mounir CHRIT (mounir.chrit@enpc.fr)

**Abstract.**

In the framework of ChArMEx (the Chemistry-Aerosol Mediterranean Experiment), the air-quality model Polyphemus is used to understand the sources of inorganic and organic particles in the western Mediterranean and to evaluate the uncertainties linked to the model parameters (meteorological fields, anthropogenic and sea-salt emissions, hypotheses related to the
model representation of condensation/evaporation). The model is evaluated by comparisons to in-situ aerosol measurements performed during three consecutive summers (2012, 2013 and 2014). The model-to-measurement comparisons concern the concentrations of $PM_{10}$, $PM_1$, Organic Matter in $PM_1$ ($OM_1$) and inorganic aerosol concentrations monitored at a remote site (Ersa) in Corsica Island, as well as during airborne measurements performed above the western Mediterranean Sea. Organic particles are mostly from biogenic origin. The model parameterization of sea-salt emissions has shown to strongly influence
the concentrations of all particulate species ($PM_{10}$, $PM_1$, $OM_1$ and inorganic concentrations). Although the emission of organic matter by the sea has shown to be low, organic concentrations are influenced by sea-salt emissions, because they provide a mass onto which gaseous hydrophilic organic compound can condense. $PM_{10}$, $PM_1$, $OM_1$ are also very sensitive to meteorology, because it affects not only the transport of pollutants, but also natural emissions (biogenic and sea salt). To avoid large and unrealistic sea-salt concentrations, a parameterization with an adequate wind-speed power law is chosen. Sulfate is
shown to be strongly influenced by anthropogenic (ship) emissions. $PM_{10}$, $PM_1$, $OM_1$ and sulfate concentrations are better described using the emission inventory with the best spatial description of ships emissions (EDGAR-HTAP). However, this is not true for nitrate, ammonium and chloride concentrations, which are very dependent on the hypotheses used in the model for condensation/evaporation. Model simulations show that sea-salt aerosols above the sea are not mixed with background transported aerosols. Taking into account the mixing state of particles with a dynamic approach of condensation/evaporation
may be necessary to accurately represent inorganic aerosol concentrations.

# 1 Introduction

Fine particulate matter (PM) in the atmosphere are of concern due to their effects on health, climate, ecosystems and biological cycles, and visibility. These effects are especially important in the Mediterranean region. The western Mediterranean basin experiences high gaseous pollution levels originating from Europe (Millán et al., 1997; Debevec et al., 2017; Doche et al., 2014; Menut et al., 2015; Nabat et al., 2013; Safieddine et al., 2014) in particular during summer, when photochemical activity is at its maximum. Furthermore, the western Mediterranean basin is impacted by various natural sources: Saharan dust, intense biogenic emissions in summer, oceanic emissions, and biomass burning, all of them being emitters of gases (e.g. volatile organic compounds (VOC), nitrogen oxides ($NO_x$)) and/or primary particles (Bossioli et al., 2016; Tyrlis and Lelieveld, 2012; Monks et al., 2009; Gerasopoulos et al., 2006). During the TRAQA 2012 and SAFMED 2013 measurement campaigns, Di Biagio et al. (2015) observed that aerosols in the western Mediterranean basin are strongly impacted by dust outflows and continental pollution. A large part of this continental pollution is secondary, i.e. it is formed in the atmosphere by chemical reactions (e.g. Sartelet et al., 2012). These reactions involve compounds, which may be emitted from different sources (e.g. biogenic and anthropogenic). Using measurements and/or modeling, several studies showed that as much as 70% to 80% of organic aerosol in summer in the western Mediterranean region is secondary and from contemporary origins (El Haddad et al., 2011; Chrit et al., 2017).

Air-quality models are powerful tools to simulate and predict the atmospheric chemical composition and the properties of aerosols at regional scales. In spite of the tremendous efforts deployed recently, the sources and the transformation mechanisms of atmospheric aerosols are not fully characterized nor fully understood. For organic aerosols, difficulties in the modeling partly lie in the representation of volatile and semi-volatile organic precursors in the models, which can only take into account a limited number of compounds or classes of compounds (Kim et al., 2011a; Chrit et al., 2017). Difficulties in modeling atmospheric particles are strongly linked to uncertainties in meteorology and in emissions (Roustan et al., 2010). For example, the turbulent vertical mixing affects the dilution and chemical processing of aerosols and their precursors (Nilsson et al., 2001; Aan de Brugh et al., 2012), clouds affect aerosol chemistry and size distribution (Fahey and Pandis, 2001; Ervens et al., 2011), and photochemistry (Tang et al., 2003; Feng et al., 2004), and precipitation controls wet deposition processes (Barth et al., 2007; Yang et al., 2012; Wang et al., 2013). Over the Mediterranean region, uncertainties due to meteorology and transport may strongly impact pollutant concentrations, because the basin is influenced by pollution transported from different regions, such as dust from Algeria, Tunisia and Morocco as well as both biogenic and anthropogenic species from Europe (Chrit et al., 2017; Denjean et al., 2016). Chrit et al. (2017) and Cholakian et al. (2018) showed that although organic aerosol concentrations at a remote marine site of the western Mediterranean are mostly of biogenic origin, they are strongly influenced by air-masses transported from the continent and by maritime shipping emissions.

In addition to the meteorological uncertainties, uncertainties in emission inventories are also important. There are uncertainties in biogenic emissions (Sartelet et al., 2012), as well as in anthropogenic emission inventories. For anthropogenic emissions, uncertainties concern not only the emissions themselves, but also the pollutants that are to be considered in the inventory and the spatial and temporal distributions of the emissions. For example, intermediate and semi volatile organic compounds are miss-

ing from emission inventories, even though they may strongly affect the formation of organic aerosols (Couvidat et al., 2012; Denier van der Gon et al., 2015). The spatial distribution of ships and harbor traffic differs depending on emission inventories. However, over the Mediterranean Sea, ships and harbor traffic emissions may strongly affect the formation of particles. Becagli et al. (2017) found experimentally that the minimum ship emission contributions to $PM_{10}$ is 11% at Lampedusa Island, and 8% at Capo Granitola on the southern coast of Sicily. Aksoyoglu et al. (2016) showed that ship emissions in the Mediterranean may contribute up to 60% of sulfate concentrations, as $SO_2$ is a major pollutant emitted from maritime transport. However, by comparison to on-road vehicles, ships emissions are still poorly characterized (Berg et al., 2012). Besides, the multiplicity of the Mediterranean sources of pollution and their interactions makes it difficult to quantify the ship contribution to aerosol concentrations.

Seas and oceans are a significant source of sea-spray aerosols (SSA). They strongly affect the formation of cloud condensation nuclei and particle concentrations. However, according to Grythe et al. (2014), sea-spray aerosols (SSA) have one of the largest uncertainties among all emissions. The modeling of sea-salt emissions is based on empiric or semi-empiric formulas. There is a tremendous amount of parameterization of the SSA emission fluxes (Grythe et al., 2014). The SSA emission parameterization of Monahan et al. (1986) is commonly used to model sea-salt emissions of coarse particles (e.g. Sartelet et al., 2012; Solazzo et al., 2017; Kim et al., 2017). However, the strong non-linearity of the source function versus wind speed (power law with an exponent of 3.41) may lead to an overestimation of emissions at high-speed regimes, as suggested by Guelle et al. (2001) and Witek et al. (2007). Many studies showed that wind speed has the dominant influence on the sea-salt emissions (Hoppel et al., 1989; Grythe et al., 2014). However, other parameterizations use different power laws with different exponents for the wind speed (e.g. 2.07 for Jaeglé et al. (2011)), and have introduced other parameters like sea-surface temperature (Schwier et al., 2017; Jaeglé et al., 2011; Sofiev et al., 2011) and the water salinity (Grythe et al., 2014). Although the influence of marine emissions on primary organic aerosols is low for the Mediterranean (Chrit et al., 2017), their influence on inorganic aerosols is not (Claeys et al., 2017).

The aim of this work is to evaluate some of the processes that strongly affect inorganic and organic aerosol concentrations in the western Mediterranean in summer (transport and emissions), how do the data/parameterizations commonly used in air-quality models affect the concentrations. To that end, sensitivity studies relative to transport (meteorology) and emissions (anthropogenic and sea salt) are performed with the air-quality model Polyphemus and are compared to measurements performed at the marine remote Ersa super-site (Cap Corsica, France) during the summer campaigns of 2012 and 2013, and to flight measurements performed above the Western Mediterranean Sea in summer (July) 2014.

This paper is structured as follows. The air-quality model Polyphemus set-up is first described for the different input datasets/parameterizations used, as well as the measurements. Second, the meteorological fields used as input to the air-quality model are evaluated. Third, the model is evaluated by comparisons to the measurements and sensitivities studies to meteorology, sea-salt emission parameterizations and anthropogenic emissions are performed to determine the main aerosol sources and sensitivities.

## 2  Simulations set-up and measured data

In order to simulate aerosol formation over the western Mediterranean, the Polair3d/Polyphemus air quality model is used, with the set-up described in Chrit et al. (2017) and summarized here. For parameters/parameterizations that are particularly attached to uncertainties (anthropogenic emissions, meteorology, sea-salt emissions, modeling of condensation/evaporation), the alternative parameters/parameterizations that are used in the sensitivity studies are also detailed for emissions and meteorology. For computational reasons, alternative parameterizations for the modeling of condensation/evaporation are only used in the comparisons to airborne measurements in section 4.4, where they are detailed.

### 2.1  Simulations set-up and alternative parameterizations

Simulations are performed over the same domains and using the same input data as in Chrit et al. (2017).

Two nested simulations are performed: one over Europe (nesting domain, horizontal resolution: $0.5° \times 0.5°$) and one over a Mediterranean domain centered around Corsica (nested domain, horizontal resolution: $0.125° \times 0.125°$), centered around the Ersa surface super-site (red point in Figure [1]). Vertically, 14 levels are used in Polair3d/Polyphemus. The heights of the cell interfaces are 0, 30, 60, 100, 150, 200, 300, 500, 750, 1000, 1500, 2400, 3500, 6000 and 12 000 m.

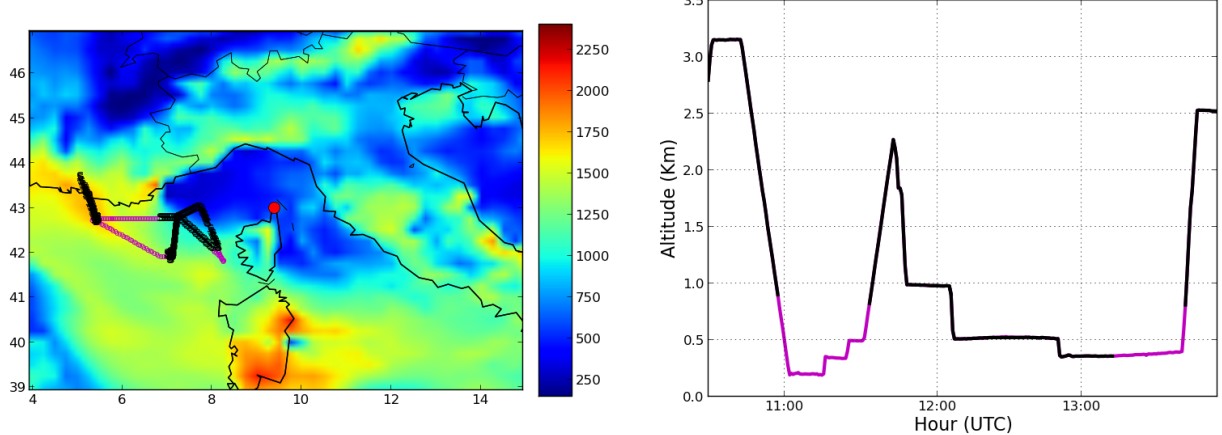

**Figure 1.** Mediterranean domain used for the simulations and planetary boundary layer (PBL) height on 10 July 2014 at noon, as obtained from the ECMWF meteorological fields (left panel). Ersa is located at the red point on northern tip of Corsica Island. The (black and purple) crosses indicate the trajectory of the flight of 10 July 2014 over the Mediterranean Sea. Altitudes during the flight (right panel). The portions conducted above the continent at the beginning and at the end of the flight from/to Avignon airport have been removed. For the model to measurement comparisons, only the transects indicated by purple crosses/lines are considered.

Simulations are performed during the summers of 2012, 2013 and 2014. The dates of simulations are chosen to match the periods of observations performed during ChArMEx (Chemistry-Aerosol Mediterranean Experiment). The Mediterranean

simulations (nested domain) are performed from 6 June to 8 July 2012, from 6 June to 10 August 2013; and from 9 to 10 July 2014. In the reference simulation, meteorological data are provided by the European Center for Medium-Range Weather Forecasts (ECMWF) model (horizontal resolution: $0.25° \times 0.25°$), which are interpolated to the Europe and Mediterranean domains. The vertical diffusion is computed using the Troen and Mahrt (1986) parameterization. In the sensitivity study rela-

tive to meteorology, meteorological fields from the Weather Research and Forecasting model (WRF, Skamarock et al. (2008)) are used in the Mediterranean simulation. WRF is forced with NCEP (National Centers for Environmental Prediction) meteorological fields for initial and boundary conditions ($1°$ horizontal grid spacing). To simulate WRF meteorological fields over the Mediterranean domain, 1-way nested WRF simulations with 24 vertical levels are conducted on two nested domains: one over Europe and one over the Mediterranean. Before conducting the sensitivity study relative to meteorology (section 3) by

using two different meteorological datasets, WRF is run with a number of different configurations, which are compared to measurements in section 3. In these configurations, the same physical parameterizations are used, but with different horizontal coordinates.

    The WRF configuration used for this study consists of the Single Moment-5 class microphysics scheme (Hong et al., 2004), the RRTM radiation scheme (Mlawer et al., 1997), the Monin-Obukhov surface layer scheme (Janjic, 2003), and the NOAA

Land Surface Model scheme for land surface physics (Chen and Dudhia, 2001). Sea surface temperature update, surface grid nudging (Liu et al., 2012; Bowden et al., 2012) options are activated.

    In the first configuration (WRF-Lon-Lat), horizontal resolutions of $0.5° \times 0.5°$ and $0.125° \times 0.125°$ are used for the nesting and nested domains respectively with a longitude-latitude projection. In the second configuration (WRF-Lambert), a Lambert (conic conform) projection is used with horizontal resolutions of 55.65 km $\times$ 55.65 km and 13.9 km $\times$ 13.9 km for the

nesting and nested domains respectively. The third configuration (WRF-Lambert-OBSGRID) also uses a Lambert projection, but the meteorological fields are improved by nudging global observations of temperature, humidity and wind from surface and radiosonde measurements (NCEP operational global surface and upper-air observation subsets, as archived by the Data Support Section (DSS) at NCAR (National Center for Atmospheric Research)).

    Biogenic emissions are estimated using Model of Emissions of Gases and Aerosols from Nature (MEGAN) with the stan-

dard MEGAN LAIv database (MEGAN-L, Guenther et al. (2006)) and the EFv2.1 dataset. For the different simulations, these emissions are recalculated with the meteorological data used for transport. In the reference simulation, yearly anthropogenic emissions are generated using the EDGAR-HTAP_V2 inventory for 2010 (http://edgar.jrc.ec.europa.eu/htap_v2/). EDGAR-HTAP_V2 inventory uses total national emissions from the European Monitoring and Evaluation Program (EMEP) emission inventory that are re-allocated spatially using EDGAR4.1 proxy subset (Janssens-Maenhout et al., 2012). The differences be-

tween the two inventories do not lie only in the spatial allocation of emissions, but also in the spatial resolution. EMEP provides a resolution of $0.5°$ x $0.5°$ while the resolution of EDGAR-HTAP_V2 is $0.1°$ x $0.1°$. To illustrate the differences between the two inventories, $NO_x$ emissions from the EMEP emission inventory, as well as absolute differences of $NO_x$ emissions between the HTAP and EMEP inventories are shown in Figure 2. The highest discrepancies between the two inventories mostly concern the shipping emissions (very low in EMEP emission inventory ($< 0.2$ $\mu$g m$^{-2}$ s$^{-1}$) whereas they can be as high as

2.8 $\mu$g m$^{-2}$ s$^{-1}$ over the sea in HTAP emission inventory), as well as emissions over large cities, mostly Genoa, Marseille

and Rome (with emissions as high as 2.5 $\mu$g.m$^{-2}$.s$^{-1}$ higher in HTAP emission inventory). HTAP emissions are used in the reference simulation and EMEP emissions for a sensitivity study as shown in Table 1.

Sea-salt emissions are parameterized using Jaeglé et al. (2011) in the reference simulation and using the commonly-used Monahan et al. (1986) for a sensitivity study. These two parameterizations are based on open-sea measurements, but they are
different in terms of the source function.

These two parameterizations are different in terms of the source function, which is defined as the total mass of sea-salt aerosol (SSA) released by area and time units. In fact, the source functions of these two parameterizations have a different dependency on the wind speed. In terms of emitted sea-salt mass, the largest differences are located over the sea in the south of France (with differences as high as 1400%), where the shear stress exerted by the wind on the sea surface is the highest.
Following Schwier et al. (2015), the emitted dry sea-salt mass is assumed to be made of 25.40% of chloride, 30.61% of sodium and 4.22% of sulfate.

The boundary conditions for the European simulation are calculated from the global model MOZART4 (Horowitz et al., 2003) (https://www.acom.ucar.edu/ wrf-chem/mozart.shtml), and those for the Mediterranean domain are obtained from the European simulation. Mineral dust emissions are not calculated in the model, but are provided from the boundaries and their
heterogeneous reactions to form nitrate and sulfate are not taken into account.

The numerical algorithms used for transport and the parameterizations used for dry and wet depositions are detailed in (Sartelet et al., 2007). Gas-phase chemistry is modeled with the carbon bond 05 mechanism (CB05) (Yarwood et al., 2005), to which reactions are added to model the formation of secondary organic aerosols (Kim et al., 2011b; Chrit et al., 2017).

The SIze REsolved Aerosol Model (SIREAM; Debry et al. (2007)) is used for simulating the dynamics of the aerosol
size distribution by coagulation and condensation/evaporation. SIREAM uses a sectional approach and the aerosol distribution is described here using 20 sections of bound diameters: 0.01, 0.0141, 0.0199, 0.0281, 0.0398, 0.0562, 0.0794, 0.1121, 0.1585, 0.2512, 0.3981, 0.6310, 1.0, 1.2589, 1.5849, 1.9953, 2.5119, 3.5481, 5.0119, 7.0795 and 10.0 $\mu$m. The condensation/evaporation of inorganic aerosols is determined using the thermodynamic model ISORROPIA (Nenes et al., 1998) with a bulk equilibrium approach in order to compute the partitioning between the gaseous and particle phases of aerosols. Because
the concentrations and the partitioning between gaseous and particle phases of chloride, nitrate and ammonium are strongly affected by condensation/evaporation and reactions with other pollutants, sensitivities of these concentrations to hypothesis used in the modeling (thermodynamic equilibrium, mixed sea-salt and anthropogenic aerosols) are also performed (section 4.4.2). For organic aerosols, the gas–particle partitioning of the surrogates is computed using SOAP assuming bulk equilibrium (Couvidat and Sartelet, 2015). The gas–particle partitioning of hydrophobic surrogates is modeled following Pankow (1994), with
absorption by the organic phase (hydrophobic surrogates). The gas– particle partitioning of hydrophilic surrogates is computed using the Henry's law modified to extrapolate infinite dilution conditions to all conditions using an aqueous-phase partitioning coefficient, with absorption by the aqueous phase (hydrophilic organics, inorganics and water). Activity coefficients are computed with the thermodynamic model UNIFAC (UNIversal Functional group; (Fredenslund et al., 1975). After condensation/evaporation, the moving diameter algorithm is used for mass redistribution among size bins. As detailed in (Chrit et al.,
2017), anthropogenic intermediate/semi-volatile organic compounds (I/S-VOC) emissions are emitted as three primary surro-

gates of different volatilities (characterized by their saturation concentrations C*: log(C*) = -0.04, 1.93, 3.5). The ageing of each primary surrogate is represented through a single oxidation step, without NOx dependence, to produce a secondary surrogate of lower volatility (log(C*) = -2.4, -0.064, 1.5 respectively) but higher molecular weight. Gaseous I/S-VOC emissions are missing from emission inventories, they are estimated here as detailed in (Zhu et al., 2016) by multiplying the primary organic emissions (POA) by 1.5, and by assigning them to species of different volatilities. A sensitivity study where I-S/VOC emissions are not taken into account is also performed.

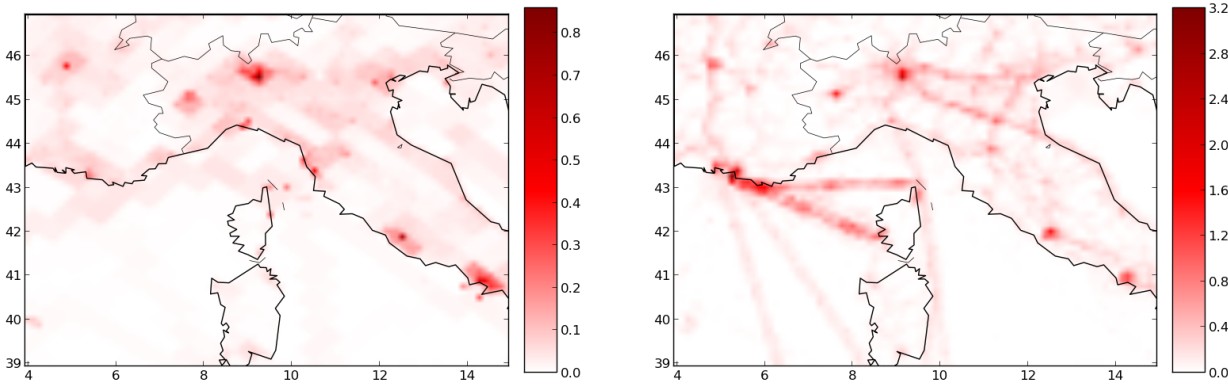

**Figure 2.** Average NO$_x$ emissions over the summer campaign 2013 from the EMEP emission inventory (left panel) and absolute differences ($\mu$g m$^{-2}$ s$^{-1}$) of NO$_x$ emissions between HTAP and EMEP inventories. The horizontal and vertical axes show longitude and latitude in degrees respectively.

Sensitivity studies to meteorology fields, anthropogenic emission inventory, I/S-VOC emissions and sea-salt emissions are outlined in section 4. These studies are performed using two different inputs for the parameter of concern in the sensitivity test and fixing the others. Table 1 summarizes the performed simulations as well as the different input data used. Table 2 summarizes the different simulation comparisons, as performed in the conducted sensitivity studies.

**Table 1.** Summary of the different simulations and their input data. S1, S2, S3, S4 and S5 represent the simulation number.

| Nomenclature | Anthropogenic emission inventory | Meteorological model | Sea-salt emission parameterization | I/S-VOC/POA |
|---|---|---|---|---|
| S1 | HTAP | ECMWF | Jaeglé et al. (2011) | 1.5 |
| S2 | HTAP | WRF Lon-Lat | Jaeglé et al. (2011) | 1.5 |
| S3 | HTAP | ECMWF | Monahan et al. (1986) | 1.5 |
| S4 | EMEP | ECMWF | Jaeglé et al. (2011) | 1.5 |
| S5 | HTAP | ECMWF | Jaeglé et al. (2011) | 0.0 |

**Table 2.** Summary of the different sensitivity simulations for the ground-based evaluation.

| Sensitivity study to | Compared simulations | Discussed concentrations | Period |
|---|---|---|---|
| Meteorology | S1 and S2 | Inorganics - $PM_{10}$ - $PM_1$ - $OM_1$ | Summer 2013 |
| Anthropogenic emission inventory | S1 and S4 | Inorganics - $PM_{10}$ - $PM_1$ - $OM_1$ | Summers 2012 and 2013 |
| Marine emissions | S1 and S3 | Inorganics - $PM_{10}$ - $PM_1$ - $OM_1$ | Summer 2013 |
| I/S-VOC/POA | S1 and S5 | $OM_1$ | Summer 2013 |

## 2.2 Measured data

The model results are compared against observational data performed in the framework of several ChArMEx campaigns. Simulated concentrations in the first vertical level of the model are compared to ground-based measurements performed at Ersa (43°00'N, 9°21.5'E), which is located at the northern edge of Corsica Island, at a height of about 530 m above sea level (Figure 1). A Campbell meteorological station was used to measure air temperature and wind speed. Continuous measurements of $PM_{10}$ and $PM_1$ were performed using TEOM (THERMO, model 1400) and TEOM-FDMS (THERMO, model 1405) instruments respectively. For the composition of particles, nitrate, sulfate, ammonium and organic concentrations in $PM_1$ were characterized using an ACSM (Aerosol Chemical Speciation Monitor), and in $PM_{10}$ they were characterized using a PILS-IC (Particle Into Liquid Sampler coupled with Ion Chromatography), which also allows an estimation of chloride and sodium concentrations (see Michoud et al. (2017) for more details). The inorganic precursors $HNO_3$, $HCl$ and $SO_2$ were measured using a WAD-IC (Wet-Annular Denuder coupled with Ion Chromatography).

Airborne measurements based in Avignon, France were preformed aboard the ATR-42, run by SAFIRE (French aircraft service for environmental research, http://safire.fr). Full details of the aerosol measurements aboard the aircraft as well as the flight details are provided in Freney et al. (2018). On 10 July 2014, a flight was dedicated to measure concentrations above the sea under Mistral regime (northern and northwestern high-speed winds). This flight was approximately 3 hours in duration and the aircraft flew over the south of France and the Mediterranean Sea at altitudes varying from 100 to 3000 m above sea level (m.a.s.l). Comparisons between model and measurements are not performed during transit, but only above sea, at altitudes below 800 m.a.s.l. and in the boundary layer. A horizontal projection of the aircraft path during this flight is presented in Figure 1. The purple crosses indicate the locations where model and measurement comparisons are performed. Measurements of the non-refractory submicron aerosol chemical properties were performed using a compact aerosol time of flight mass spectrometer (C-ToF-AMS) providing mass concentrations on organic sulfate, ammonia and chloride particles with a time resolution of less than 5 minutes.

## 3 Meteorological evaluation

Aerosol phenomenology in Cape-Corsica is influenced by diverse meteorological situations as well as transport of pollution from a number of sources. It is therefore crucial to estimate, as accurately as possible, the input meteorological data used in the air quality model. The four meteorological datasets (ECMWF, WRF-Lon-Lat, WRF-Lambert, WRF-Lambert-Obsgrid) are compared to observations of air temperature and wind at Ersa in Figure B1 for the summer campaign periods of 2012 and Figure B2 for the summer 2013 (Appendix B).

The observed and simulated temperature, wind speed, wind direction and relative humidity at Ersa during these summers, and the statistical scores defined in Table [A1] of Appendix A and comparison of the four model results to measurements (hourly time series) are shown in Tables 3 to 6 respectively.

**Table 3.** Temperature (observed and simulated means) from the observations and the four meteorological models at Ersa during the summer campaigns 2012 and 2013 and statistics of comparison of model results to observations (correlation, mean fractional bias, mean fractional error, mean bias and gross error). The temperature means and the RMSE are in Kelvin. $\bar{o}$ refers to the measured mean.

| Meteorological models | | ECMWF | WRF-Lon-Lat | WRF-Lambert | WRF-Lambert-OBSGRID |
|---|---|---|---|---|---|
| **2012** $\bar{o} = 294.66$ | Simulated mean $\bar{s} \pm$ RMSE | 295.09 ± 1.50 | 294.05 ± 2.79 | 294.86 ±3.02 | 294.17 ± 3.45 |
| | Correlation (%) | 96.3 | 77.1 | 66.7 | 54.8 |
| | MFB | 0.00 | 0.00 | 0.00 | 0.00 |
| | MFE | 0.00 | 0.01 | 0.01 | 0.01 |
| | MB | 0.43 | -0.61 | 0.20 | -0.49 |
| | GE | 1.33 | 2.38 | 2.56 | 2.93 |
| **2013** $\bar{o} = 294.04$ | Simulated mean $\bar{s} \pm$ RMSE | 295.82 ± 3.23 | 294.42 ± 2.42 | 295.31 ± 2.66 | 295.10 ± 2.60 |
| | Correlation (%) | 70.0 | 78.2 | 79.0 | 78.3 |
| | MFB | 0.01 | 0.00 | 0.00 | 0.00 |
| | MFE | 0.01 | 0.01 | 0.01 | 0.01 |
| | MB | 1.79 | 0.38 | 1.27 | 1.06 |
| | GE | 2.69 | 2.01 | 2.17 | 2.14 |

As mentioned in EPA 2007 report, Emery et al. (2001) proposed benchmarks for temperature (mean bias (MB) within ±0.5 K and gross error (GE) 2.0 K), wind speed (MB within ±0.5 m.s$^{-1}$ and RMSE < 2 m.s$^{-1}$ ) and wind direction (MB within ±10° and GE < 30°). McNally (2009) suggested an alternative set of benchmarks for temperature (MB within ±1.0 K and GE < 3.0 K).

The four meteorological simulations reproduce well the ground temperature measured at Ersa. Although only the ECMWF temperature in 2012 verifies the US EPA criteria, all simulations verify the criterion of McNally (2009) for the GE. Statistically, the correlation to temperature measurements is high: between about 54% and 96% for all models, and the root-mean-square-error (RMSE) is low (below 3.4 K). The best model differs depending on the year: the correlation of ECMWF to measurements is the highest (96%) and the RMSE the lowest (1.5 K) in 2012, but in 2013, the correlation of ECMWF is the lowest (70%)

and its RMSE the highest (3.2 K). The mean fractional biases and errors (MFB and MBE) of the simulated temperatures are almost null.

    For wind speed, ECMWF systematically leads to better statistics than WRF, despite the fine horizontal resolution of WRF (0.125° x 0.125°). ECMWF agrees best with the measurements, with the highest correlation (between 69% and 87%) and the lowest errors (MFE is between 33% and 47%). It also verifies the US EPA criteria for both the summers 2012 and 2013.

WRF-Lon-Lat also performs well with correlations between 60% and 65% and MFE between 47% and 64%. WRF-Lambert and WRF-Lambert-Obsgrid have poorer statistics with negative correlations and MFE between 71% and 74%.

    The averaged wind direction is quite similar for summers 2012 and 2013 (202° and 186° respectively). The mean wind direction is best represented by ECMWF for the summers 2012 and 2013, and WRF-Lon-Lat for the summer 2012. However,

**Table 4.** Wind speed statistics for the four meteorological models at Ersa during the summer campaigns of 2012 and 2013. The wind speed means and the RMSE are in m.s$^{-1}$. $\bar{o}$ refers to the measured mean.

| Meteorological models | | ECMWF | WRF-Lon-Lat | WRF-Lambert | WRF-Lambert-OBSGRID |
|---|---|---|---|---|---|
| 2012 $\bar{o} = 4.53$ | Simulated mean $\bar{s} \pm$ RMSE | 4.86 ± 2.36 | 6.96 ± 3.93 | 5.60 ± 3.94 | 5.06 ± 3.89 |
| | Correlation (%) | 69.3 | 60.3 | -26.0 | -34.3 |
| | MFB | 0.14 | 0.46 | 0.34 | 0.26 |
| | MFE | 0.47 | 0.64 | 0.74 | 0.74 |
| | MB | 0.33 | 2.40 | 1.07 | 0.52 |
| | GE | 1.89 | 3.28 | 3.45 | 3.34 |
| 2013 $\bar{o} = 3.21$ | Simulated mean $\bar{s} \pm$ RMSE | 3.44 ± 1.32 | 3.98 ± 2.12 | 5.14 ± 3.64 | 4.86 ± 3.44 |
| | Correlation (%) | 87.3 | 65.5 | -6.6 | -2.1 |
| | MFB | 0.01 | 0.10 | 0.38 | 0.30 |
| | MFE | 0.33 | 0.47 | 0.73 | 0.71 |
| | MB | -0.35 | 0.19 | 1.36 | 1.07 |
| | GE | 1.01 | 1.59 | 3.06 | 2.88 |

**Table 5.** Wind direction statistics for the four meteorological models at Ersa during the summer campaigns 2012 and 2013. The wind direction means and the RMSE are in degrees. $\bar{o}$ refers to the measured mean.

| Meteorological models | | ECMWF | WRF-Lon-Lat | WRF-Lambert | WRF-Lambert-OBSGRID |
|---|---|---|---|---|---|
| 2012 $\bar{o} = 201.89$ | Simulated mean $\bar{s} \pm$ RMSE | 195.73 ± 91.64 | 200.48 ± 58.94 | 107.07 ± 120.47 | 101.30 ± 119.53 |
| | Correlation (%) | 27.6 | 54.1 | 7.2 | 12.0 |
| | MFB | -0.14 | -0.02 | -0.62 | -0.66 |
| | MFE | 0.40 | 0.22 | 0.68 | 0.69 |
| | MB | -6.16 | -1.41 | -94.82 | -100.59 |
| | GE | 62.09 | 39.74 | 104.00 | 104.43 |
| 2013 $\bar{o} = 186.28$ | Simulated mean $\bar{s} \pm$ RMSE | 206.67 ± 107.84 | 231.03 ±117.91 | 101.57 ± 120.47 | 111.46 ± 122.76 |
| | Correlation (%) | 33.2 | 21.6 | 3.6 | 1.7 |
| | MFB | -0.02 | 0.13 | -0.50 | -0.48 |
| | MFE | 0.48 | 0.46 | 0.67 | 0.68 |
| | MB | 20.38 | 44.74 | -84.71 | -74.83 |
| | GE | 73.96 | 81.34 | 100.13 | 101.88 |

**Table 6.** Relative humidity statistics for the four meteorological models at Ersa during the summers 2012 and 2013. The relative humidity means and the RMSE are dimensionless. $\bar{o}$ refers to the measured mean.

| Meteorological models | | | ECMWF | WRF-Lon-Lat | WRF-Lambert | WRF-Lambert-OBSGRID |
|---|---|---|---|---|---|---|
| 2012 $\bar{o} = 0.65$ | | Simulated mean $\bar{s} \pm$ RMSE | $0.74 \pm 0.24$ | $0.72 \pm 0.22$ | $0.70 \pm 0.25$ | $0.77 \pm 0.25$ |
| | | Correlation (%) | 14.3 | 34.5 | 7.9 | 14.0 |
| | | MFB | 18 | 15 | 11 | 31 |
| | | MFE | 32 | 28 | 32 | 31 |
| | | MB | 0.09 | 0.07 | 0.05 | 0.12 |
| | | GE | 0.20 | 0.18 | 0.20 | 0.20 |
| 2013 $\bar{o} = 0.70$ | | Simulated mean $\bar{s} \pm$ RMSE | $0.73 \pm 0.20$ | $0.78 \pm 0.21$ | $0.70 \pm 0.20$ | $0.69 \pm 0.21$ |
| | | Correlation (%) | 9.7 | 23.3 | 23.0 | 21.8 |
| | | MFB | 8 | 14 | 3 | 1 |
| | | MFE | 26 | 25 | 25 | 25 |
| | | MB | 0.17 | 0.17 | 0.17 | 0.17 |
| | | GE | 0.03 | 0.08 | 0.00 | -0.01 |

the modeled wind speed does not respect the US EPA criteria. Errors are higher with the two models using Lambert projection which tend to under-estimate the wind direction angle. For relative humidity, the observed mean relative humidity is 0.65 in 2012 and 0.70 in 2013. It is relatively well reproduced by the models (between 0.70 and 0.77 in 2012 and between 0.69 and 0.78 in 2013). All models perform well with MFE below 32% and MFB below 18%. WRF-Lon-Lat leads to the best statistics

in 2012 and WRF-Lambert-Obsgrid leads to the best statistics in 2013.

As ECMWF and WRF-Lon-Lat are shown to perform overall better than the two other models (Tables 3, 4, 6, 5), they will be used for the meteorological sensitivity study.

The model performances presented above compare well to other studies (Kim et al., 2013; Cholakian et al., 2018). In this study, for ECMWF and WRF-Lon-Lat during the summers of 2012 and 2013, RMSE ranges between 1.5 K and 3.2 K for

temperature, between 1.3 m s$^{-1}$ and 3.9 m s$^{-1}$ for wind speed, and between 58° and 118° for wind direction. At Ersa, for the summer 2013 (not exactly the same period), Cholakian et al. (2018) found RMSE between 1.5 K and 2.3 K for temperature, between 1.6 m s$^{-1}$ and 1.9 m s$^{-1}$ for wind speed, and between 92° and 117° for wind direction at ERSA from 10 July to 5 August 2013 using the mesoscale WRF model. Moreover, Kim et al. (2013) reported RMSE ranging between 1 K and 4 K for temperature, and 0.6 m s$^{-1}$ to 3.0 m s$^{-1}$ for wind speed over Greater Paris during May 2005 using WRF model with a

longitude-latitude map projection.

## 4 Evaluation and sensitivities

This section focuses on the evaluation of the reference simulation (S1) against aerosol measurements ($PM_{10}$, $PM_1$, $OM_1$ and inorganic aerosols (IA) species), and on the factors controlling simulated aerosol concentrations (meteorology, sea-salt and anthropogenic emissions). This evaluation is performed against ground-based measurements during the summers 2012 and 2013, and against airborne measurements during the flight of 10 July 2014. The criteria of Boylan and Russell (2006) are used to evaluate the model-to-measurement comparisons. The performance criterion is verified if $|MFB| \leq 60\%$ and $MFE \leq 75\%$ (MFB and MFE stand respectively for the mean fractional bias and the mean fractional error and are defined in Table [A1] of Appendix A), while the goal criterion is verified if $|MFB| \leq 30\%$ and $MFE \leq 50\%$. To evaluate the sensitivity of the modeled concentrations to input data, the different simulations summarized in Table 1 are compared to the reference simulation S1 by computing the normalized root-mean-square-error (RMSE of the concentration differences between a simulation and S1, divided by the mean concentration of S1).

### 4.1 $PM_{10}$ and $PM_1$

The statistical scores of the simulated $PM_1$ and $PM_{10}$ are shown in Table 7 for the summer campaigns of 2012 and 2013. The time series of measured and simulated $PM_{10}$ and $PM_1$ during the summer 2013 are presented in Figure C1 of Appendix C.

$PM_{10}$ and $PM_1$ are well modeled during both the summer campaigns of 2012 and 2013, and the performance and goal criteria are always met. The measured mean concentration of $PM_1$ is very similar in 2012 and 2013 (7.6 and 7.0 $\mu$g m$^{-3}$ respectively). However, the mean $PM_{10}$ concentration is double in 2012 compared to 2013 (22.4 and 11.5 $\mu$g m$^{-3}$ respectively), probably because of higher occurrence of transported desert dust in 2012 (Nabat et al., 2015).

Although the mean $PM_1$ and $PM_{10}$ concentrations are well modeled in 2013, the mean $PM_1$ concentration is slightly under-estimated during summer 2013 and the mean $PM_{10}$ concentration is slightly under-estimated in 2012. This under-estimation of $PM_{10}$ may be due to difficulties in accurately representing the transported dust episodes, which are frequent in summer in the western Mediterranean (Moulin et al., 1998) and are represented in the Mediterranean simulation by dust boundary conditions from the global model MOZART4.

The comparisons of the different simulations at Ersa in Table 7 shows that both $PM_{10}$ and $PM_1$ concentrations are strongly influenced by sea-salt emissions (S3, with a normalized RMSE of 65% and 40% respectively), especially as the emissions of the two parameterizations differ by as much as 1400% over the sea in Southern France (section 2.1). $PM_{10}$ and $PM_1$ concentrations are also very sensitive to meteorology (S2, with a normalized RMSE of 33% and 21% respectively) and anthropogenic emissions (S4, with a normalized RMSE of 17% and 10% respectively).

Knowing the chemical composition of $PM_{10}$ and $PM_1$ provides important information to understand the different sources of aerosol particles arriving at Ersa, and to understand the sensitivities presented above. Figure 3 shows the simulated composition of $PM_{10}$ and $PM_1$ and the percentage contribution of each compound to PM in 2013 and the associated variability.

According to simulation, inorganic aerosols account for a large part of the $PM_{10}$ mass: during the summer campaign periods of 2012 and 2013, the inorganic fraction in $PM_{10}$ is 31% and 39% respectively. Among inorganics, sulfate, largely originating

**Table 7.** Comparisons of simulated $PM_{10}$, $PM_1$ and $OM_1$ daily concentrations to observations (concentrations and RMSE are in $\mu g\ m^{-3}$) during the summer campaign periods of 2012 (between 09 June and 03 July) and 2013 (between 07 June and 03 August). $\bar{s}$ stands for simulated mean, and $\bar{o}$ for observed mean. Simulation details are given in table 1

| | | $PM_{10}$ (2012) | $PM_1$ (2012) | $OM_1$ (2012) | $PM_{10}$ (2013) | $PM_1$ (2013) | $OM_1$ (2013) |
|---|---|---|---|---|---|---|---|
| | Measured mean $\bar{o}$ | 22.38 | 7.57 | 3.89 | 11.46 | 7.02 | 2.88 |
| **S1** | $\bar{s} \pm$ RMSE | 16.44 ± 7.55 | 9.40 ± 2.72 | 3.39 ± 0.78 | 9.69 ± 3.17 | 6.98 ± 1.77 | 2.56 ±1.07 |
| | Correlation (%) | 76.8 | 78.9 | 95.2 | 70.9 | 67.5 | 81 |
| | MFB | -30 | 18 | -20 | -19 | -1 | -17 |
| | MFE | 30 | 27 | 23 | 26 | 20 | 35 |
| **S2** | $\bar{s} \pm$ RMSE | — | — | — | 7.49 ± 4.75 | 6.42 ± 1.91 | 1.61 ± 1.62 |
| | Diff. with S1 (%) | — | — | — | -23 | -8 | -37 |
| | Norm. RMSE (%) | — | — | — | 33 | 21 | 49 |
| **S3** | $\bar{s} \pm$ RMSE | — | — | — | 14.94 ± 5.02 | 9.45 ± 2.95 | 3.26 ± 1.03 |
| | Diff. with S1 (%) | — | — | — | 54 | 35 | 27 |
| | Norm. RMSE (%) | — | — | — | 65 | 40 | 29 |
| **S4** | $\bar{s} \pm$ RMSE | 13.87 ± 10.95 | 7.66 ± 1.56 | 2.37 ± 1.64 | 8.48 ± 4.02 | 6.86 ± 2.03 | 1.98 ± 1.29 |
| | Diff. with S1 (%) | -16 | -19 | -30 | -12 | -2 | -23 |
| | Norm. RMSE (%) | 23 | 2 | 43 | 17 | 10 | 32 |
| **S5** | $\bar{s} \pm$ RMSE | — | — | — | — | — | 2.54 ± 1.07 |
| | Diff. with S1 (%) | — | — | — | — | — | -1 |
| | Norm. RMSE (%) | — | — | — | — | — | 1 |

from anthropogenic sources, occupies a large portion of $PM_{10}$ (18% in 2012 and 19% in 2013). The organic mass (OM) also largely contributes to $PM_{10}$ (30% in 2012 and 33% in 2013). Black carbon (originating from traffic and shipping emissions and industrial activities in big cities in the south of France and the north of Italy) contributes to a small portion of $PM_{10}$ (5% in 2012 and 7% in 2013). Saharan dust can be transported by air-masses to the Mediterranean atmosphere via medium-range transport and is an important component of the $PM_{10}$ with contributions of 34% and 21% during the summer campaigns of 2012 and 2013, respectively.

The $PM_1$ mass is dominated by organic matter (41% in 2012 and 38% in 2013) and sulfate (30% in 2012 and 24% in 2013). The percentage of sodium (from sea salt) is significant in $PM_{10}$ (4% in 2012 and 10% in 2013), but it is negligible in the $PM_1$ mass (less than 1%).

## 4.2 $OM_1$

The statistical evaluation of $OM_1$ during the summer campaigns of 2012 and 2013 is available in Table 7. As discussed in Chrit et al. (2017), the performance and goal criteria are both satisfied, due to the addition in the model of highly oxidized species (ex-

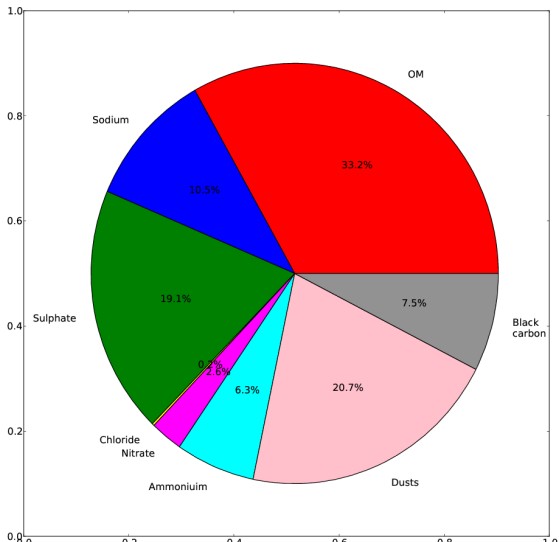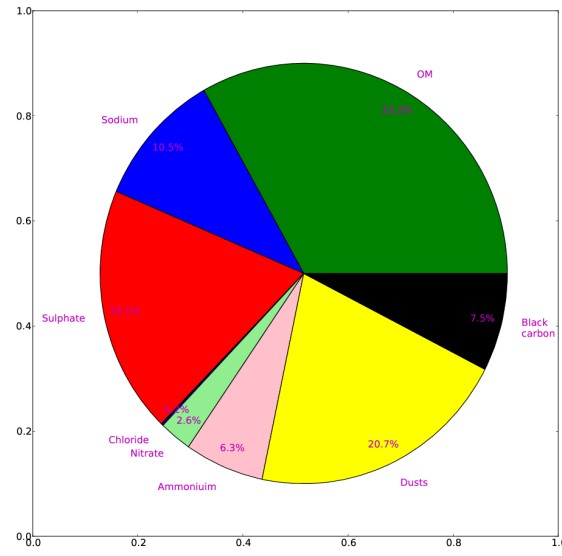

**Figure 3.** PM$_{10}$ (left panel) and PM$_1$ (right panel) average relative simulated composition during the summer 2013 campaign period

tremely low volatility organic compounds, organic nitrate and the carboxylic acid MBTCA (3-methyl-1,2,3-butanetricarboxylic acid) as a second generation oxidation product of $\alpha$-pinene). Adding these species to the model was also required to correctly model OM properties (oxidation state and affinity to water). The time series of measured and simulated OM$_1$ concentrations during the summer 2013 campaign are presented in Figure C1 of Appendix C. The comparison of the different simulations at Ersa in Table 7 shows that OM$_1$ is particularly influenced by meteorology (S2 with a normalized RMSE of 49%), because meteorology influences biogenic emissions, but also by inorganic sea-salt emissions (S3 with a normalized RMSE of 29%), which provides mass onto which hydrophilic SOA can condense especially sulfate, and anthropogenic emissions (S4 with a normalized RMSE of 32%), as they affect the formation of oxidants through photochemistry and emit anthropogenic precursors. The sensitivity to anthropogenic I/S-VOC emissions is low (S5, with a normalized RMSE of only 1%).

## 4.3 Inorganic species

### 4.3.1 Ground-based evaluation

The statistical scores of the simulated inorganic concentrations are shown in Table 8 for PM$_1$ concentrations during the summer 2012 campaign and in Table 9 and Table 10 for PM$_{10}$ and PM$_1$ inorganic concentrations respectively during the summer 2013 campaign. The time series of measured and simulated inorganic concentrations during the summer 2013 campaign are presented in Figures C2 and C3 of Appendix C.

Inorganic concentrations of $PM_1$ aerosol were measured in 2012, and in both $PM_1$ and $PM_{10}$ in 2013. Some of the inorganic gaseous precursors ($SO_2$, $HNO_3$ and $HCl$) were also measured only for a few days in 2013 (between 21 July and 26 July 2013).

For the 2012 reference simulation (S1), for the $PM_1$, sulfate and nitrate concentrations satisfy both the performance and goal criteria. Ammonium concentrations are however under-estimated, despite the performance criterion being satisfied in terms of MFE. This under-estimation of ammonium increases if EMEP emission inventory with lower ship emissions over the Mediterranean Sea is used, suggesting that ammonium nitrate formation is strongly dependent on the ship $NO_x$ emissions (because they lead to the formation of the gaseous precursors $HNO_3$ of ammonium nitrate).

For the 2013 reference simulation (S1), in $PM_{10}$, sulfate and ammonium satisfy the performance and goal criteria, while sodium satisfies only the performance criterion. The mean concentrations of modeled chloride and nitrate are both under-estimated. This under-estimation is probably due to uncertainties on the measurements. In fact, nitrate and chloride are difficult to measure, there can be some negative artifacts (volatilization of the aerosol phase during sampling) or positive artefact's (condensation of gaseous phase onto the particles or filters during sampling), depending on the sampling conditions. Moreover, this underestimation may be also due to uncertainties on the modeled temperature (with bias as high as about 5 K in daily points), and due to difficulties in representing the partitioning between the gas and the particle phases. For chloride, as shown in Figure C2 of Appendix C, although the mean concentration is under-estimated, the peaks are over-estimated. For example, between 21 and 26 July 2013, the particle-phase chloride concentration is 0.34 $\mu$g m$^{-3}$ in the simulation, but only 0.05 $\mu$g m$^{-3}$ in the measurements. The total chloride (gas + particle phase) is well modeled (1.2 $\mu$g m$^{-3}$ in the measurements and 1 $\mu$g m$^{-3}$ simulated), but the gas/particle ratio is much higher in the measurements (18.4) than in the model (2.4). For nitrate, the total nitrate (gas + particle phase) is over-estimated between 21 and 26 July 2013 (2.7 $\mu$g m$^{-3}$ in the measurements and 6.6 $\mu$g m$^{-3}$ simulated), and most of it is in the gas phase (only 0.4 $\mu$g m$^{-3}$ in the particle phase in the measurements and 0.2 simulated). Contrary to chloride, the gas/particle ratio is much higher in the model (28.2) than in the measurements (5.4). The reason for these difficulties to represent the gas/particle ratios of chloride is that the measured PILS chloride concentrations include only non-refractory chloride. The reason of the difference of nitrate ratio is likely related to the internal-mixing hypothesis and the bulk-equilibrium assumption in the modeling of condensation/evaporation. They are investigated in the following section, during the comparison to airborne measurements.

For the 2013 reference simulation (S1), in $PM_1$, as in $PM_{10}$, sulfate and ammonium satisfy the performance criterion, which is also almost satisfied for nitrate. The measured and simulated $PM_1$ and $PM_{10}$ concentrations are relatively similar for sulfate and ammonium, suggesting that most of the mass is in $PM_1$.

The comparisons of the different simulations at Ersa in Tables 9 and 10 show that inorganics in $PM_{10}$ and $PM_1$ have similar sensitivities, because of the bulk equilibrium assumption made in the modeling of condensation/evaporation. Sulfate is more sensitive to anthropogenic (ship) emissions (with a normalized RMSE of 44% in $PM_{10}$) than meteorology (with a normalized RMSE of 22%) and sea-salt emissions (with a normalized RMSE of 22%). Nitrate, chloride and sodium, and ammonium to a lower extent, are highly sensitive to sea-salt emissions with normalized RMSEs between 62% and 933% (Jaeglé et al. (2011) parameterization has a lower dependance to wind speed than Monahan et al. (1986) parameterization). They are also strongly affected by meteorology (with normalized RMSEs between 43% and 130%), because meteorology

**Table 8.** Comparisons of simulated PM$_1$ inorganic daily concentrations to observations (concentrations are in $\mu$g m$^{-3}$) using S1 and S4 during the summer 2012.

| | Inorganics | Nitrate | Sulfate | Ammonium | |
|---|---|---|---|---|---|
| | Measured mean $\bar{o}$ | 0.41 | 2.06 | 1.39 | |
| **S1** | Simulated mean $\bar{s} \pm$ RMSE | 0.51 ±0.28 | 2.53 ± 1.13 | 0.68 ±0.85 | 0.04 ± 0.03 |
| | Correlation (%) | 20.1 | 71.4 | 47.8 | |
| | MFB | 15 | 31 | -72 | |
| | MFE | 50 | 39 | 72 | |
| **S4** | Simulated mean $\bar{s} \pm$ RMSE | 0.53 ± 0.36 | 1.71± 1.28 | 0.50 ± 1.04 | |
| | Diff. with S1 (%) | +4% | -32% | -26% | |
| | Norm. RMSE (%) | 45 | 46 | 32 | |

affects the natural emissions (sea salt and biogenic), as discussed in section 5. By influencing biogenic emissions, meteorology affects the formation of organics (Sartelet et al., 2012), because they are mostly of biogenic origins in summer (Chrit et al., 2017). The influence of meteorology on biogenic emissions also affects the formation of inorganics because of the modification of oxidant concentrations (Aksoyoglu et al., 2017) and because of the temperature bias that can be as high as 5K and the formation of organic nitrate (Ng et al., 2017). Inorganic concentrations are also strongly affected by anthropogenic emissions (with normalized RMSEs between 44% and 267%), because anthropogenic emissions affect the NO$_x$ emissions, and hence the oxidants and both organic and inorganic nitrate formation. Because nitrate, ammonium and chloride partition between the gas and the particle phases, their uncertainties are linked and they are strongly affected by assumptions in the modeling of condensation/evaporation, as detailed in the section 4.4.

## 4.4 Airborne evaluation

The considered flight (10 July 2014, 10:21-14:09 UTC) was conducted by the French aircraft ATR42 deployed by SAFIRE in the south of France above the Mediterranean Sea. The purpose was to study aerosol formation, evolution and properties in marine conditions, under Mistral regime (north/north-west winds coming from the Rhône Valley characterized by high wind speeds). Altitudes and a horizontal projection of the trajectory of the aircraft during this flight are presented in Figure 1. The aircraft flew at low altitudes (under 800 m.a.s.l.) over the Mediterranean Sea for about 2 hours, allowing us to evaluate the modeling of sea-salt aerosols. As shown in Figure 1, the planetary boundary layer height, as modeled by ECMWF meteorological fields, exhibit strong spatial variations.

For the comparisons of inorganic concentrations to airborne measurements, the reference simulation S1 is run a few days during the summer 2014. The simulated concentrations are extracted along the flight path from the corresponding grid cells and layers. For the model to measurement comparisons, only the cells were the plane was flying above the sea, at low altitudes (below 800 m.a.s.l.) with a spatially uniform boundary layer (above 1200 m) are considered. The transects where model to measurements are performed are indicated by purple crosses/lines in Figure 1. The meteorological fields during this flight are

**Table 9.** Comparisons of simulated $PM_{10}$ inorganic daily concentrations to observations (concentrations are in $\mu g\ m^{-3}$) using S1, S2, S3 and S4 during the summer 2013.

| | Inorganics | Nitrate | Sulfate | Ammonium | Chloride | Sodium |
|---|---|---|---|---|---|---|
| | Measured mean $\bar{o}$ | 0.42 | 1.52 | 0.76 | 0.18 | 0.53 |
| S1 | Simulated mean $\bar{s} \pm$ RMSE | 0.33 ± 0.42 | 2.05 ± 0.84 | 0.58 ± 0.39 | 0.12 ± 0.45 | 0.70 ± 0.54 |
| | Correlation (%) | 5.7 | 69.7 | 47.6 | -11.4 | 55.5 |
| | MFB | -43 | 32 | -20 | -67 | 30 |
| | MFE | 86 | 40 | 43 | 105 | 70 |
| S2 | Simulated mean $\bar{s} \pm$ RMSE | 0.19 ±0.46 | 2.10 ± 0.82 | 0.49 ± 0.44 | 0.13 ± 0.44 | 0.77 ±0.57 |
| | Diff. with S1 (%) | -42% | +2 % | -16% | +8% | +10% |
| | Norm. RMSE (%) | 130 | 22 | 52 | 100 | 43 |
| S3 | Simulated mean $\bar{s} \pm$ RMSE | 0.88 ± 1.27 | 2.14 ±0.97 | 0.31 ± 0.60 | 0.59 ± 1.14 | 1.77 ± 2.34 |
| | Diff with S1 (%) | +167% | +4% | -47% | +392% | +153% |
| | Norm. RMSE (%) | 376 | 22 | 62 | 933 | 291 |
| S4 | Simulated mean $\bar{s} \pm$ RMSE | 0.24 ±0.41 | 1.33 ±0.67 | 0.34 ± 0.56 | 0.27 ± 0.64 | 0.98 ± 0.77 |
| | Diff. with S1 (%) | -27% | -35% | -41% | +125% | +40% |
| | Norm. RMSE (%) | 66 | 44 | 48 | 267 | 50 |

**Table 10.** Comparisons of simulated $PM_1$ inorganic daily concentrations to observations (concentrations are in $\mu g\ m^{-3}$) using S1, S2, S3 and S4 during the summer 2013.

| | Inorganics | Nitrate | Sulfate | Ammonium |
|---|---|---|---|---|
| | Measured mean $\bar{o}$ | 0.30 | 1.47 | 0.65 |
| S1 | Simulated mean $\bar{s} \pm$ RMSE | 0.32 ± 0.31 | 1.86 ±0.94 | 0.58 ± 0.38 |
| | Correlation (%) | 22.9 | 28.9 | 32 |
| | MFB | -24 | 27 | -6 |
| | MFE | 77 | 55 | 55 |
| S2 | Simulated mean $\bar{s} \pm$ RMSE | 0.18 ± 0.28 | 1.72 ± 0.66 | 0.50 ± 0.52 |
| | Diff. with S1 (%) | -44% | -8% | -14% |
| | Norm. RMSE (%) | 134 | 19 | 44 |
| S3 | Simulated mean $\bar{s} \pm$ RMSE | 0.87 ± 1.20 | 1.89 ± 0.81 | 0.31 ± 0.50 |
| | Diff. with S1 (%) | +172% | +2% | -47% |
| | Norm. RMSE (%) | 384 | 29 | 62 |
| S4 | Simulated mean $\bar{s} \pm$ RMSE | 0.23 ± 0.25 | 1.08 ± 0.71 | 0.34 ±0.48 |
| | Diff. with S1 (%) | -28% | -42% | -41% |
| | Norm. RMSE (%) | 69 | 34 | 47 |

compared with measured data in Appendix F. The Mistral regime is simulated with wind directions that are well modeled, although wind speeds are underestimated.

### 4.4.1 Sulfate

Figure 4 shows the comparison of sulfate to the airborne measurements using different model configurations. Sulfate is the
inorganic compound with the highest $PM_1$ concentrations (about 0.54 $\mu$g m$^{-3}$).

As shown in Figure 4, the $PM_1$ sulfate concentration is over-estimated in the simulation with a mean concentration of about 0.55 $\mu$g m$^{-3}$ against 0.47 $\mu$g m$^{-3}$ in the measurements. To understand the reasons of this over-estimation, different sensitivity simulations are performed. The first sensitivity simulation (referred to as "S1-without-$SO_4$ in SSE", where SSE stands for sea-salt emissions) differs from S1 simulation by the fact that sulfate is only emitted from anthropogenic sources and marine
sulfate is not taken into account. The second sensitivity simulation (referred to as "S1-$H_2SO_4$-0%") differs from S1 by the fact that $SO_x$ emissions are split into 100% of $SO_2$ and 0% of $H_2SO_4$, instead of 98% of $SO_2$ and 2% of $H_2SO_4$ in S1. The measurement-to-model comparison of the vertical profile of the $PM_1$ sulfate concentrations using the three simulations is shown in Figure 4. The influence of marine sulfate is negligible: the simulated means using S1 with and without the emissions of marine sulfate are nearly equal ($\approx$ 0.55 $\mu$g.m$^{-3}$) indicating that the $PM_1$ sulfate concentration is almost totally from
anthropogenic sources. A comparison of $PM_{10}$ sulfate concentrations for the two simulations show that this is also the case for $PM_{10}$ sulfate concentrations. This is indicative of the overestimation of sulfate or sulfuric acid emissions, or to the treatment in the model of emissions from ship stacks. However, $PM_1$ sulfate concentrations are strongly influenced by anthropogenic emissions. For example, $PM_1$ sulfate concentrations are lower if the fraction of $H_2SO_4$ in the $SO_x$ emissions is lower than in the reference simulation (the simulated mean concentrations with and without $H_2SO_4$ in $SO_x$ emissions are 0.55 and 0.52 $\mu$g.m$^{-3}$
respectively), because of the rapid condensation of $H_2SO_4$ (which saturation vapor pressure is almost zero) onto particles.

### 4.4.2 Ammonium and nitrate

Figure 5 shows the comparison of nitrate and ammonium concentrations in PM1. The simulated means of ammonium and nitrate are about 0.32 $\mu$g.m$^{-3}$ and about 0.14 $\mu$g.m$^{-3}$ respectively. In the reference simulation S1, ammonium and nitrate are under-estimated compared to the measurements.
Figure 5 shows the comparison of nitrate and ammonium concentrations in $PM_1$ to the airborne measurements using different model configurations. Because ammonium, nitrate and chloride are semi-volatile inorganic species, their concentrations may depend on the assumptions made in the modeling of condensation/evaporation. In the reference simulation, bulk thermodynamic equilibrium is assumed between the gas and particle phases for all inorganic species. In the first sensitivity simulation (referred to as "S1-Dynamic"), the condensation/evaporation is computed dynamically rather than assuming thermodynamic
equilibrium. In the second sensitivity simulation (referred to as "S1-IA-externally-mixed"), sea-salt (chloride and sodium) emissions are assumed not to be mixed with the other aerosols. In S1-IA-externally-mixed, bulk equilibrium is assumed for ammonium, nitrate and sulfate, while chloride and sodium do not interact with the other inorganic species.

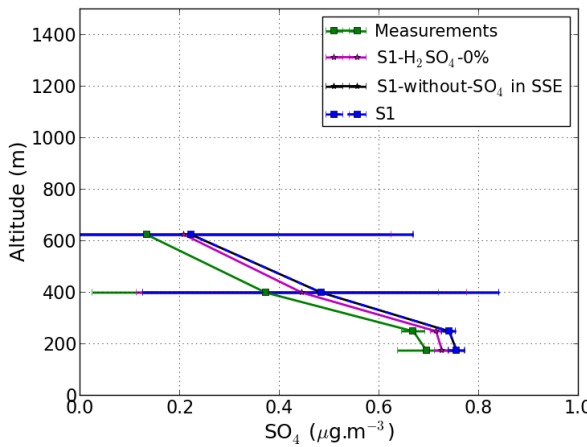

**Figure 4.** Measurements are averaged at four model levels from airborne observations below 800 m.a.g.l along the flight path shown in Figure 1 on July 10, 2014. The concentrations of the S1 simulations (standard and with options, see text for details) are also averaged in time along the flight path. Results from S1 and from S1-without-SO4 in SSE (sea-salt emissions) are quite similar.

Under the thermodynamic equilibrium approach (S1), nitrate is underestimated (the measured and simulated means are 0.10 and 0.05 $\mu$g.m$^{-3}$ respectively), probably because the sulfate is overestimated as detailed in section 4.4.1, but also because the assumption of thermodynamic equilibrium between the gas and particle phases is not verified. Nitrate concentrations are closer to measurements if condensation/evaporation is computed dynamically, especially between 400 m and 600 m altitude, where

the mean concentrations are 0.07 $\mu$g.m$^{-3}$ in the measurements, 0.02 $\mu$g.m$^{-3}$ with S1 and 0.07 $\mu$g.m$^{-3}$ with S1-Dynamic). If sea-salt aerosols are externally mixed, than nitrate is even more under-estimated than in S1. This is because nitrate tends to replace chloride in sea salt if thermodynamic consideration is taken into account.

For ammonium, the comparisons to the measurements are best if sea-salt particles are assumed not to be mixed (the measured and simulated means are 0.27 and 0.26 $\mu$g.m$^{-3}$ respectively). The differences of the vertical profiles between the dynamic

and the equilibrium approaches indicates that the assumption of the thermodynamic equilibrium is not verified (the condensation/evaporation process is not instantaneous). For instance, the simulated mean of ammonium using the equilibrium and dynamic approaches is 0.20 and 0.13 $\mu$g.m$^{-3}$ respectively.

Because both the mixing-state of particles and the dynamic of condensation/evaporation strongly influence PM$_1$ inorganic concentrations over the Mediterranean Sea, a model capable of representing the mixing state of particles with the dynamic of

condensation/evaporation (e.g. Zhu et al., 2015) may allow a better representation of inorganic concentrations.

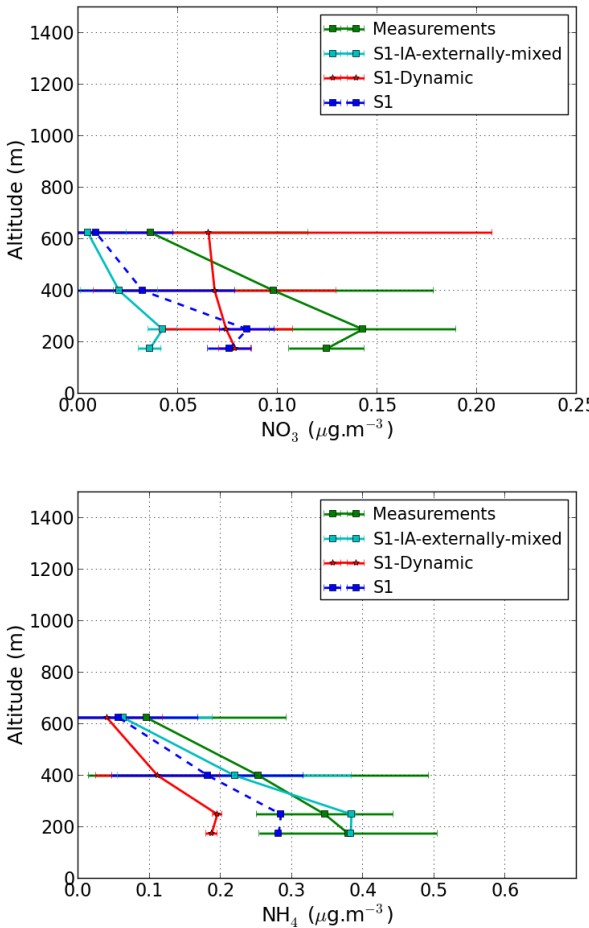

**Figure 5.** Vertical profile averaged at four model levels of $NO_3$ (left panel) and $NH_4$ (right panel). Measurements are averaged at the same four model levels from airborne observations below 800 m.a.g.l along the flight shown in Figure 1 on July 10, 2014 (around noon)

## 5 Sensitivity studies over the western Mediterranean region

Section 4 was dedicated to explain how the simulated concentrations of particles at Ersa are influenced by the different input data used (meteorology, sea salt, anthropogenic emissions) and modeling hypotheses. This section generalizes the sensitivity study of section 4 by investigating over the Mediterranean domain how the concentrations are influenced by the input data.

5    Figure D1 of Appendix D shows maps over the Mediterranean domain of the concentrations of $PM_{10}$, $OM_1$, sulfate and other secondary inorganic aerosols (nitrate, ammonium and chloride) from the simulation S1 during the summer 2013. The highest $PM_{10}$ concentrations correspond to high $OM_1$, sulfate or ammonium, nitrate, chloride concentrations. $OM_1$ concentrations are high nearby locations of high biogenic emissions such as Italy and Corsica (Figure E1 of Appendix E). Sulfate concentrations

are particularly high over the Mediterranean Sea, nearby the main ship routes (Figure 2). Ammonium nitrate concentrations are high in places of high anthropogenic emissions, such as North of Italy, as well as in main cities. Hereafter, the term VIA (Volatile Inorganic Aerosol) is used to refer to chloride, ammonium and nitrate aerosols.

Figure D2 of Appendix D shows maps of the relative difference of the concentrations of $PM_{10}$, $OM_1$, sulfate and VIA between S2 and S1 (sensitivity to meteorology). VIA concentrations show the highest sensitivity to meteorology, with relative concentration differences between S2 and S1 reaching between -90% and -60% locally over Italy. Sulfate shows the lowest sensitivity with relative concentration differences mostly between -20% and 20%. The larger influence of meteorology on VIA than on sulfate concentrations is partly explained by the influence of the temperature on the partitioning of VIA between the gas and particle phases, as VIA is highly semi volatile. $OM_1$ concentrations are quite sensitive to meteorology over the whole Mediterranean domain, with relative concentration differences mostly between -60% and -20%, especially nearby regions where the biogenic emissions are the highest. The places of the highest $OM_1$ concentrations also correspond to the places where VIA concentrations are the most sensitive to meteorology. By influencing biogenic emissions, meteorology influences the formation of organics ($OM_1$) and hence the formation of VIA by the formation of organic nitrate for example. The influence of meteorology on sulfate concentrations is limited in this study, because the formation of organo-sulfate is not modeled in our simulations.

Figure D3 of Appendix D shows maps of the relative difference of the concentrations of $PM_{10}$, $OM_1$, sulfate and VIA between S3 and S1 (sensitivity to sea-salt emissions).

As sulfate is assumed to make only 4% of sea-salt emissions (section 2.1), the influence of sea-salt emissions on sulfate concentrations at Ersa is low (the relative concentration difference is between 0% and 20%). The effect is stronger over the western part of the Mediterranean domain (with relative concentration differences between S3 and S1 between 20% and 60%), where sea-salt emissions are stronger. Chloride concentrations are also strongly influenced by sea-salt emissions, as it is directly emitted (it is assumed to make 25% of sea-salt emissions). Nitrate and ammonium concentrations are also strongly influenced by sea-salt emissions, because of thermodynamic exchanges between the gas and particle phases of chloride, nitrate and ammonium.

The influence of sea-salt emissions on $OM_1$ concentrations is also important, but it is less important than VIA (the relative concentration differences of VIA are between 90% and 180%) over the western Mediterranean part of the domain, compared to between 20% and 60% for $OM_1$ and between 40% and 60% for sulfate. The increase of $OM_1$ concentrations when sea-salt emissions are high is due to the hydrophilic organic compounds in $OM_1$, which are absorbed onto inorganic concentrations. The organic concentrations originating from sea-salt emissions are very low, as discussed in Chrit et al. (2017), and they are not taken into account here.

Figure D4 of Appendix D shows maps of the relative difference of the concentrations of $PM_{10}$, $OM_1$, sulfate and VIA between S4 and S1 (sensitivity to anthropogenic emissions). Sensitivities to sulfate and VIA concentrations are more spatially localized than sensitivities to $OM_1$ concentrations, and they are higher, with relative concentration differences between S4 and S1 between -40% and 20% for $OM_1$ and between -40% and 60% for VIA. Sulfate concentrations are strongly sensitive to anthropogenic emissions nearby the main ship routes, with negative (S4-S1) concentrations between -60% and -40%, as ship

routes are not well represented in the EMEP emission inventory (simulation S4). For VIA concentrations, the influence of anthropogenic emissions can either be negative or positive (increase or decrease of concentrations), because of the different spatial distributions of the two emission inventories, which affect directly the nitrate formation.

## 6 Conclusions

This work presents a sensitivity study to different input data and model parameterizations to better understand aerosol sources over the Mediterranean and the parameters influencing the aerosol concentrations. Aerosol sources are different depending on the aerosol chemical compounds. Comparisons to observations are performed at the Ersa station to estimate how realistic the concentrations simulated with the different parameters are (meteorological fields, anthropogenic and marine emissions, intermediate/semi-volatile organic compounds (I/S-VOC) emissions, different options in condensation/evaporation modeling).

For most pollutants, the best model performance is obtained when the meteorological fields that represent the best wind direction are used together with the emission inventory with the most accurate spatial description of ships emissions (EDGAR-HTAP).

Using ECMWF and WRF to model the meteorological fields, there is a high sensitivity of secondary pollutants (inorganics and organics) to meteorology, showing the importance of accurate meteorological modeling and the potential strong influence

of climate change on the concentrations of these secondary pollutants.

This influence of meteorology on concentrations is due to the impact on sea salt and biogenic emissions, influencing directly the formation of ammonium, nitrate, chloride and OM, as well as the impact on temperature, humidity and radiation, influencing the secondary aerosol formation. Sulfate is less sensitive to meteorology than volatile inorganic aerosols (VIA), because it is not volatile. However, this low sensitivity may change if the formation of organo-sulfate is modeled (not done here). Both inorganic

and organic concentrations are highly sensitive to sea-salt emissions, as great discrepancies exist between different published parameterizations. The commonly used Monahan parameterization of sea-salt emissions leads to an over-estimation of all particulate concentrations, especially of sodium concentrations. A parameterization with a lower exponent in the wind-speed power law is chosen to model sea-salt emissions (Jaeglé et al., 2011) and leads to better model performance. The overestimation of the modeled sea-salt concentrations using Monahan parameterization has an incidence on the overestimation of the modeled

concentrations of inorganic compounds, such as nitrate, which replaces chloride in the particles when the thermodynamic equilibrium approach is used to model condensation/evaporation. This assumption (the thermodynamic equilibrium approach) was shown not to be accurate both at Ersa and over the Mediterranean Sea. At Ersa, the gas/particle ratio is too high for nitrate and too low for chloride if the thermodynamic equilibrium approach is used, as the exchange between the gas and particle phases is not instantaneous, but it is dynamic. This dynamic exchange is strongly influenced by the particle composition,

and comparisons to measurements over the Mediterranean Sea suggest that sea-salt particles are not mixed with background (transported) particles. Overall, secondary pollutants, such as nitrate, ammonium and chloride in the particle-phase are strongly influenced by the gas/particle phase partitioning, because a high percentage of their concentrations is in the gas phase. This underlines the need to develop aerosol models able to represent accurately this gas-phase partitioning.

Sulfate originates mostly from maritime traffic. The shipping emissions lead to the formation of oxidants that in turn enhance the formation of biogenic aerosols, with the potential formation of organic nitrate and organo-sulfate. Organics are mostly from a biogenic origins during summer. Even if the contribution of sea-salt emissions to organic concentrations is low, organic concentrations are strongly influenced by sea-salt emissions, because they partition between the gas and particle phases and

5  they are hydrophilic. This underlines the need to better characterize the properties (affinity with water) of secondary organic aerosols. The emissions of I/S-VOC play a limited role in the $OM_1$ concentrations during the summer 2013, suggesting that the influence of ship emissions on $OM_1$ is mostly due to anthropogenic VOC precursors (aromatics) and $NO_x$ emissions, which lead to the formation of oxidants that may oxidize biogenic aerosol precursors (and form organic nitrate for example).

*Acknowledgements.* This research has received funding from the French National Research Agency (ANR) projects SAF-MED (grant ANR-

10  12-BS06-0013). This work is part of the ChArMEx project supported by ADEME, CEA, CNRS-INSU and Météo-France through the multidisciplinary program MISTRALS (Mediterranean Integrated Studies aT Regional And Local Scales). The station at Ersa was partly supported by the CORSiCA project funded by the Collectivité Territoriale de Corse through the Fonds Européen de Développement Régional of the European Operational Program 2007-2013 and the Contrat de Plan Etat-Région. Eric Hamounou is acknowledged for his great help in organizing the campaigns at Ersa. CEREA is a member of Institut Pierre-Simon Laplace (IPSL).

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

# Appendix A: Statistical indicators

**Table A1.** Definitions of the statistics used in this work. $(o_i)_i$ and $(c_i)_i$ are the observed and the simulated concentrations at time and location i, respectively. $n$ is the number of data

| Statistic indicator | Definition |
| --- | --- |
| Root mean square error (RMSE) | $\sqrt{\frac{1}{n}\sum_{i=1}^{n}(c_i - o_i)^2}$ |
| Correlation (Corr) | $\frac{\sum_{i=1}^{n}(c_i - \bar{c})(o_i - \bar{o})}{\sqrt{\sum_{i=1}^{n}(c_i - \bar{c})^2}\sqrt{\sum_{i=1}^{n}(o_i - \bar{o})^2}}$ |
| Mean fractional bias (MFB) | $\frac{1}{n}\sum_{i=1}^{n}\frac{c_i - o_i}{(c_i + o_i)/2}$ |
| Mean fractional error (MFE) | $\frac{1}{n}\sum_{i=1}^{n}\frac{|c_i - o_i|}{(c_i + o_i)/2}$ |
| Mean bias (MB) | $\frac{1}{n}\sum_{i=1}^{n}c_i - o_i$ |
| Gross error (GE) | $\frac{1}{n}\sum_{i=1}^{n}|c_i - o_i|$ |

## Appendix B: Meteorological evaluation

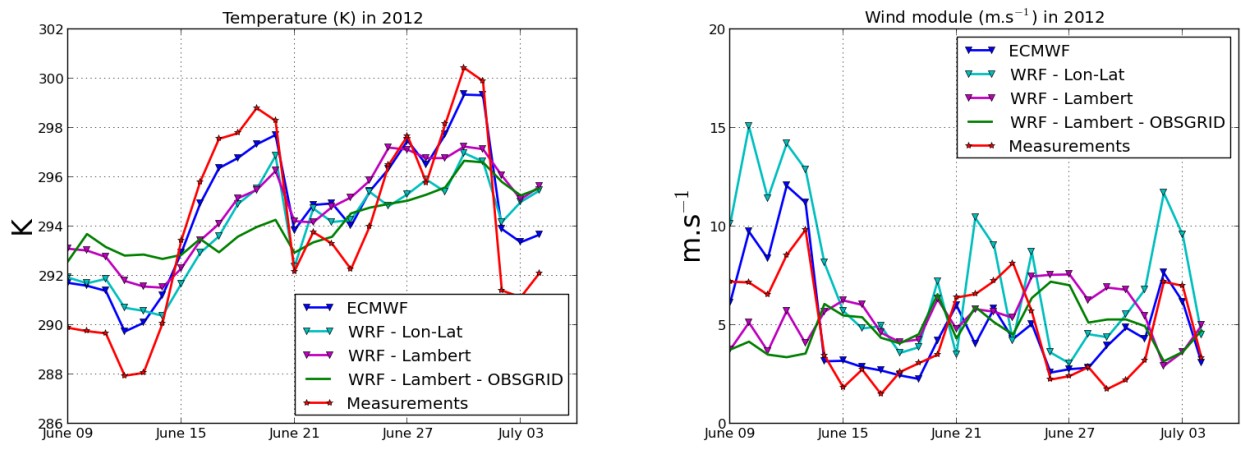

**Figure B1.** Ground Temperature (left panel) and wind speed (right panel) at Ersa during the summer 2012.

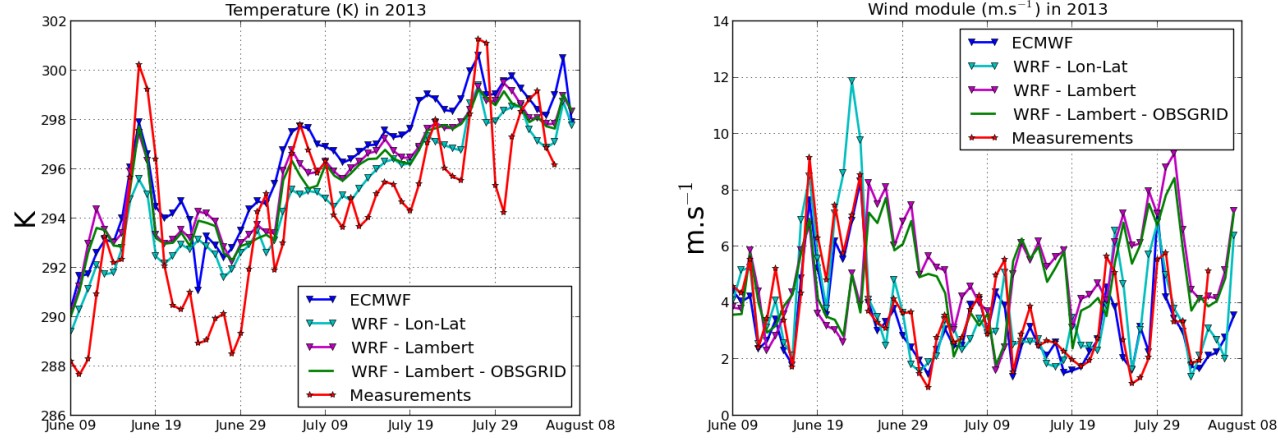

**Figure B2.** Ground Temperature (left panel) and wind speed (right panel) at Ersa during the summer 2013.

**Appendix C: Model to measurement comparisons in 2013**

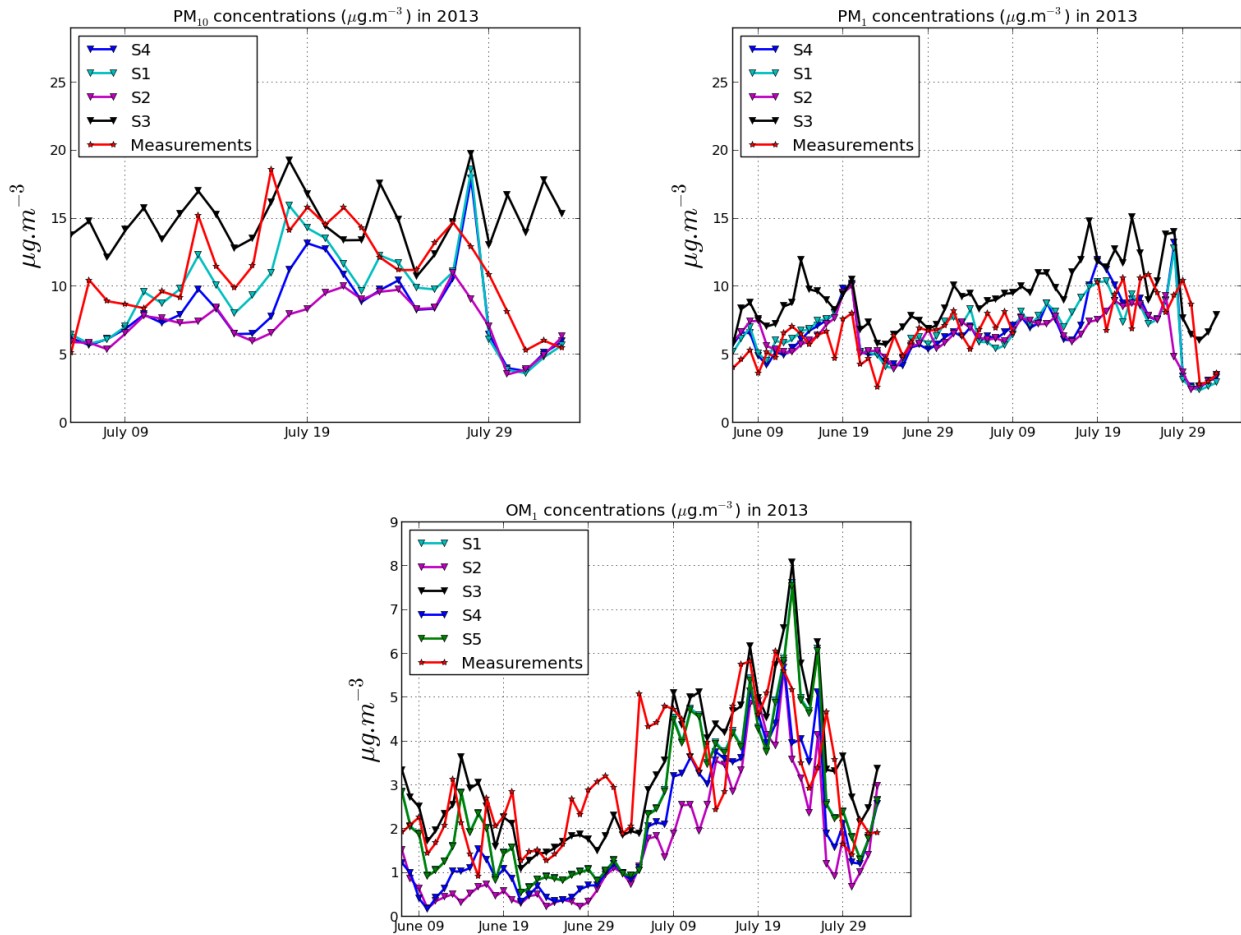

**Figure C1.** Comparisons of PM$_{10}$ (upper left panel), PM$_1$ (upper right panel), OM$_{PM_1}$ (lower panel) concentrations simulated and observed at Ersa during the summer 2013.

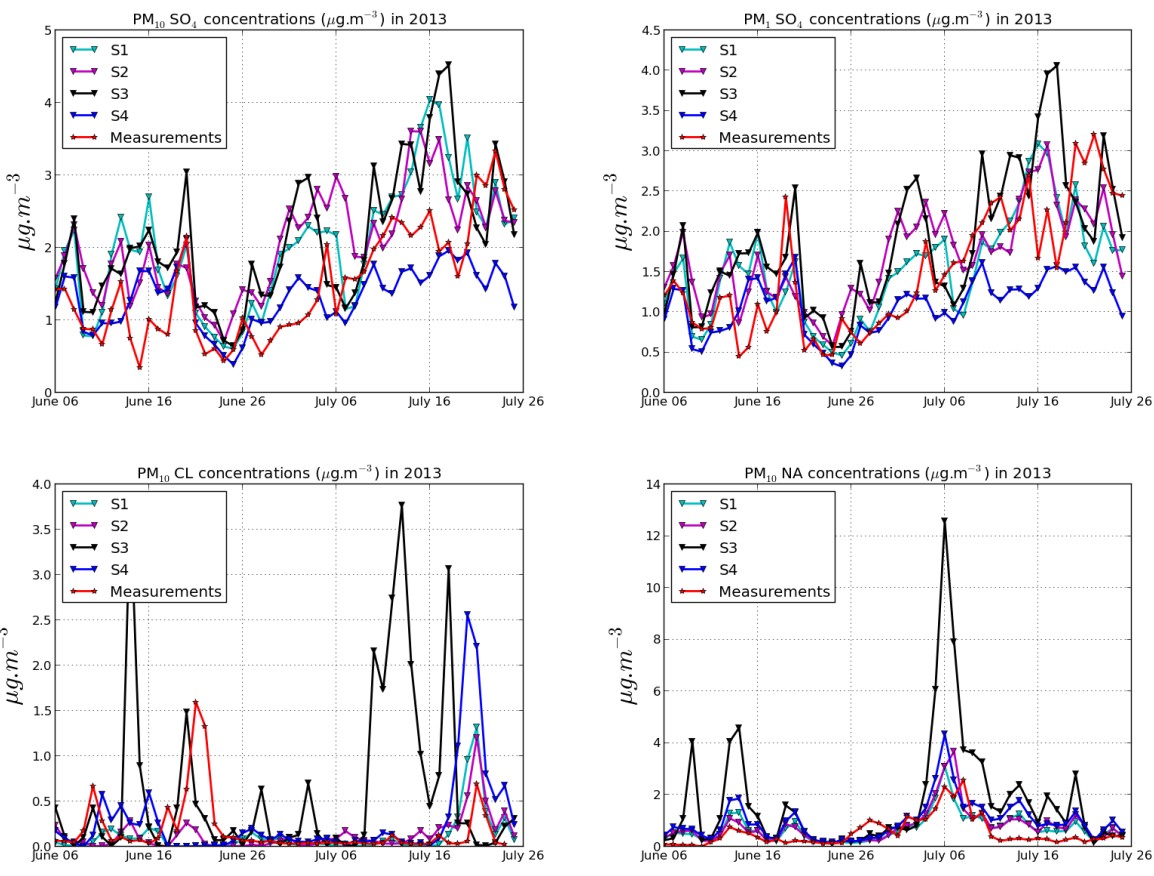

**Figure C2.** Comparisons of simulated and observed PM$_{10}$ sulfate (upper left panel), PM$_1$ sulfate (upper right panel), PM$_{10}$ chloride (lower left panel) and PM$_{10}$ sodium (lower right panel) concentrations at Ersa during the summer 2013.

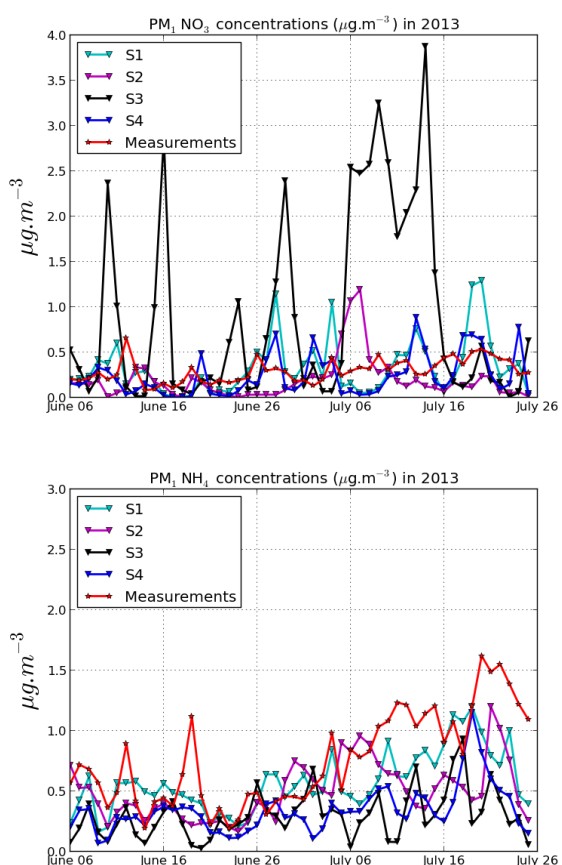

**Figure C3.** Comparisons of simulated and observed PM$_{10}$ nitrate (upper left panel), PM$_1$ nitrate (upper right panel), PM$_{10}$ ammonium (lower left panel) and PM$_1$ ammonium (lower right panel) concentrations at Ersa during the summer 2013.

**Appendix D: Concentration sensitivities in the summer 2013**

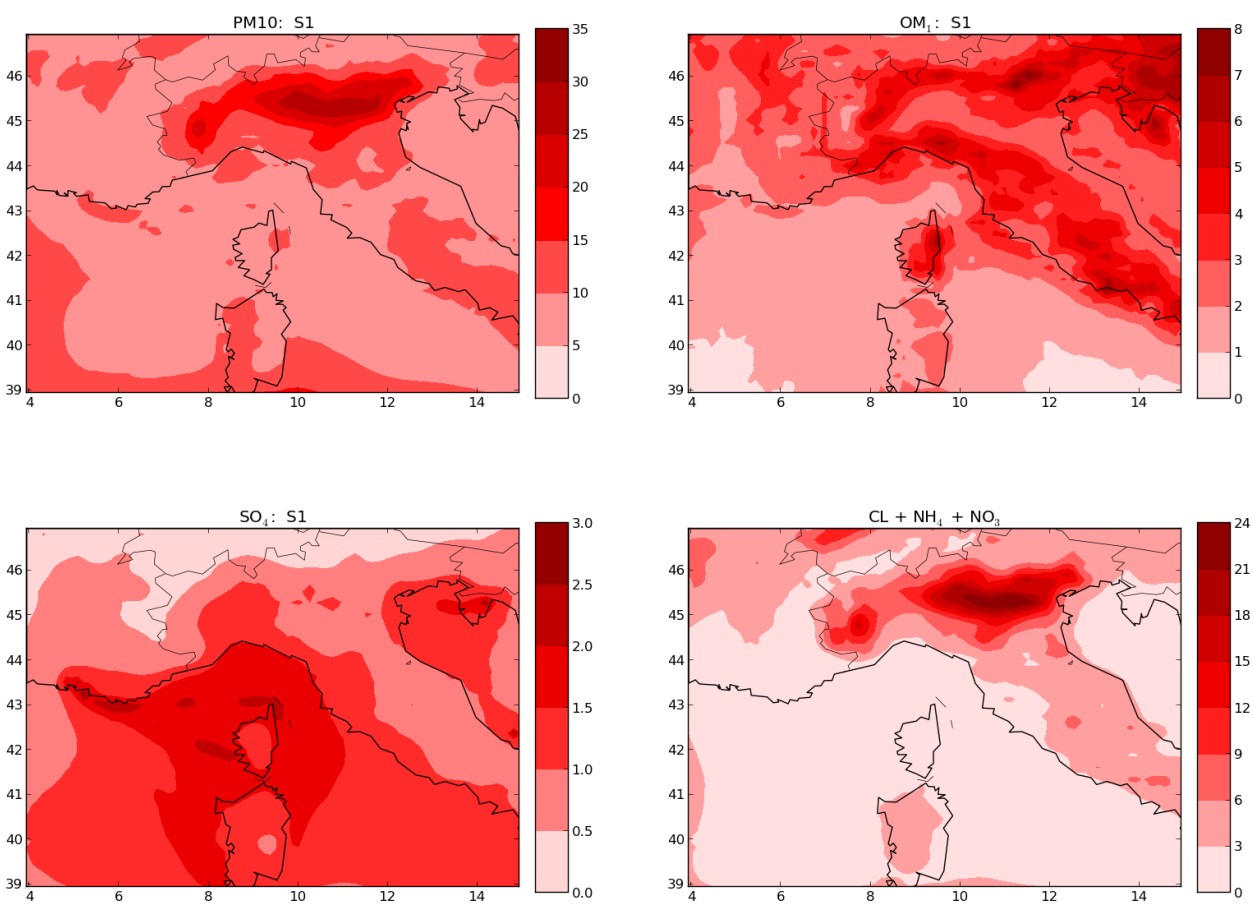

**Figure D1.** Maps of the concentrations of $PM_{10}$ (upper left panel), $OM_1$ (upper right panel), $PM_{10}$ sulfate (lower left panel) and other $PM_{10}$ inorganics (nitrate + ammonium + chloride) (lower right panel) during the summer 2013 in $\mu g\ m^{-3}$.

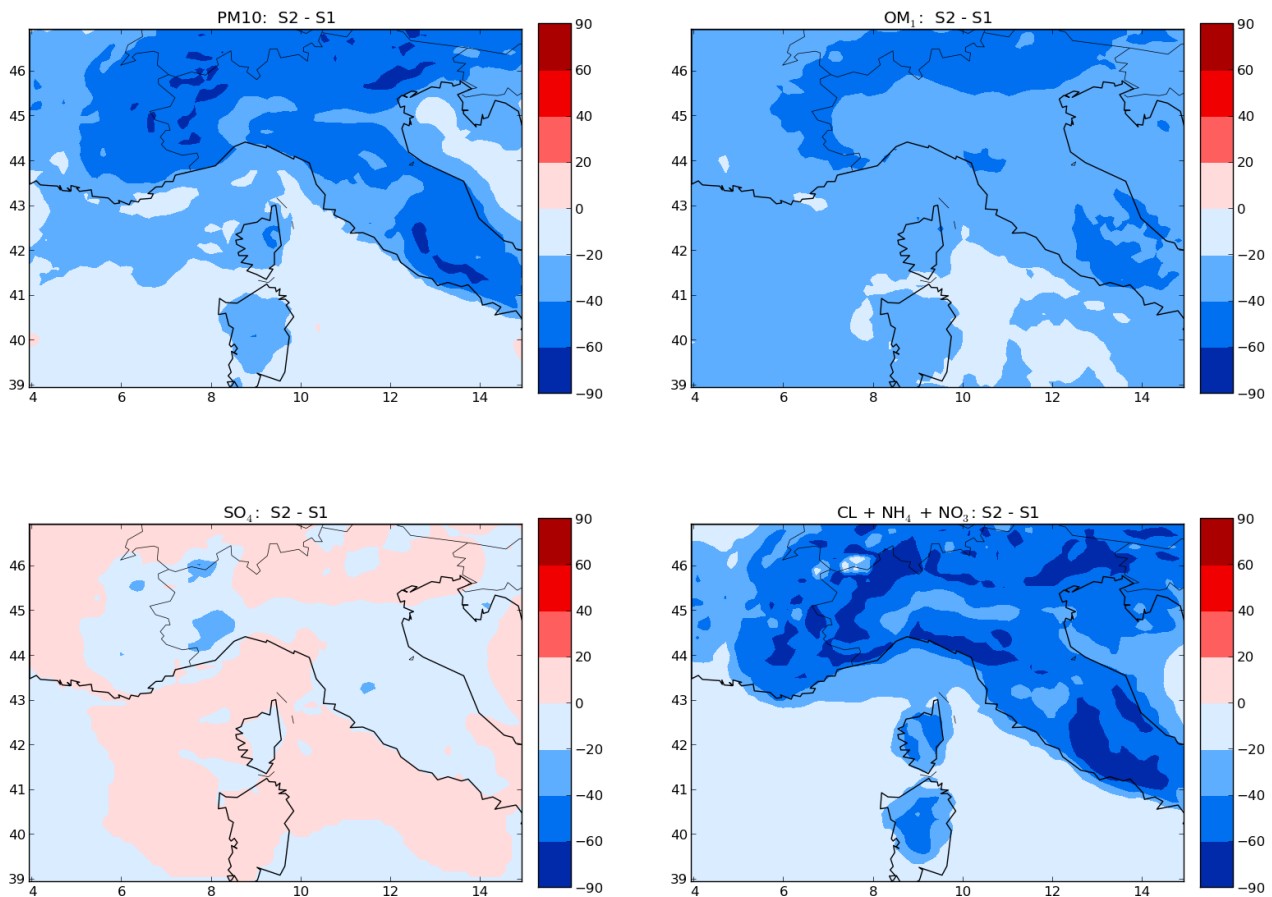

**Figure D2.** Maps of the relative differences of the concentrations of $PM_{10}$ (upper left panel), $OM_1$ (upper right panel), sulfate (lower left panel) and other inorganics (nitrate, ammonium and chloride) (lower right panel) in % between S1 and S2 (right panel) during the summer 2013.

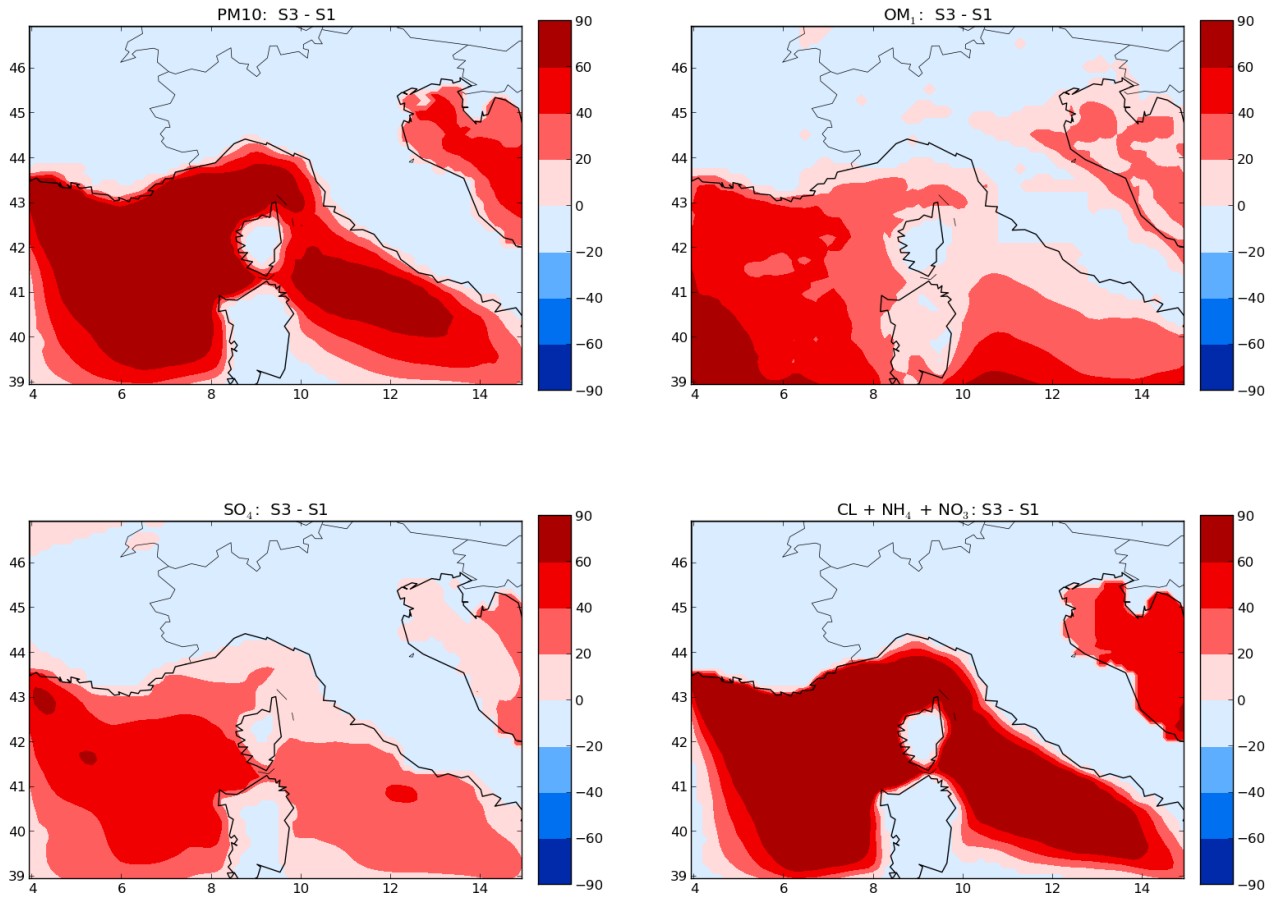

**Figure D3.** Maps of the relative differences of the concentrations of $PM_{10}$ (upper left panel), $OM_1$ (upper right panel), sulfate (lower left panel) and other inorganics (nitrate, ammonium and chloride) (lower right panel) in % between S1 and S3 (right panel) during the summer 2013.

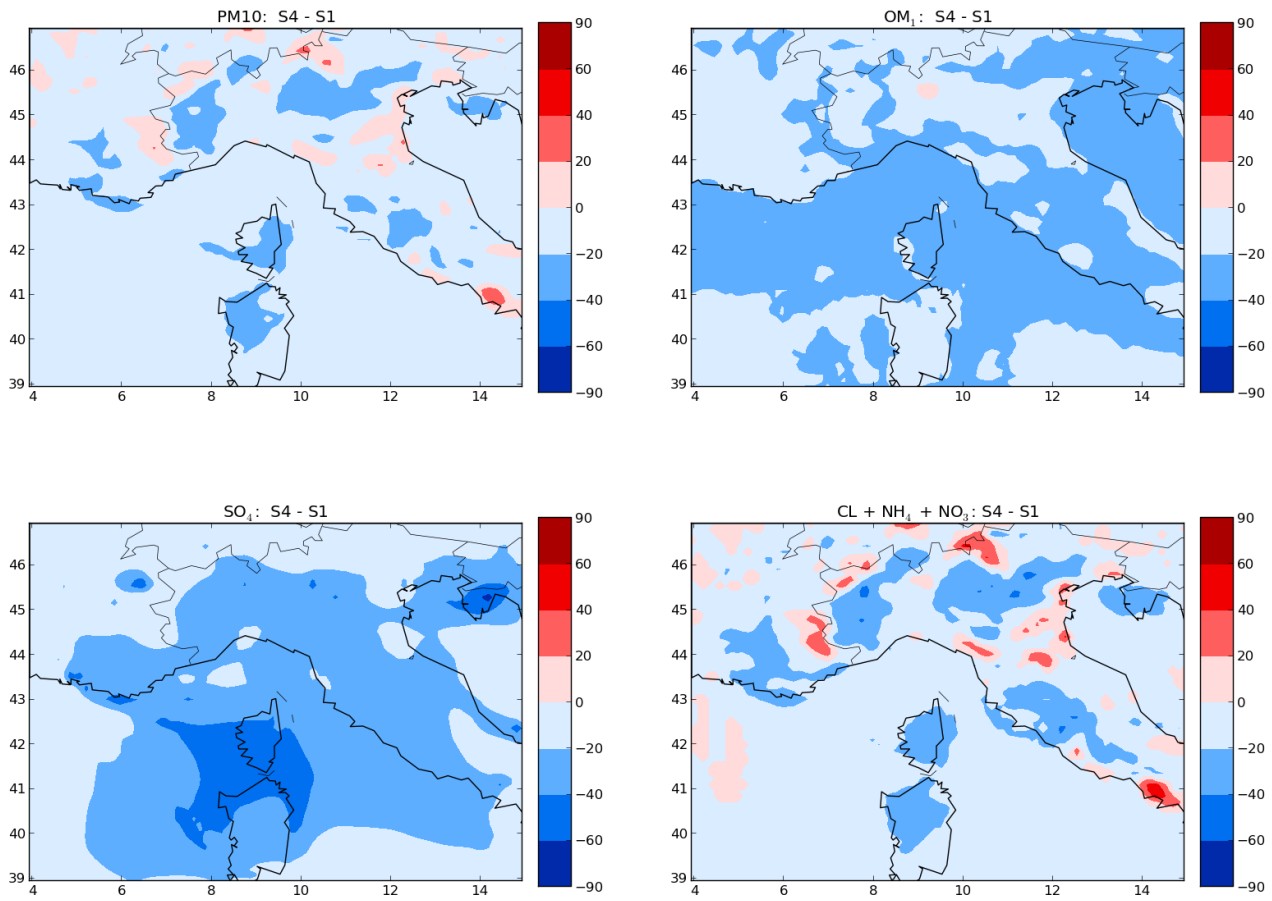

**Figure D4.** Maps of the relative differences of the concentrations of $PM_{10}$ (upper left panel), $OM_1$ (upper right panel), sulfate (lower left panel) and other inorganics (nitrate, ammonium and chloride) (lower right panel) in % between S1 and S4 (right panel) during the summer 2013.

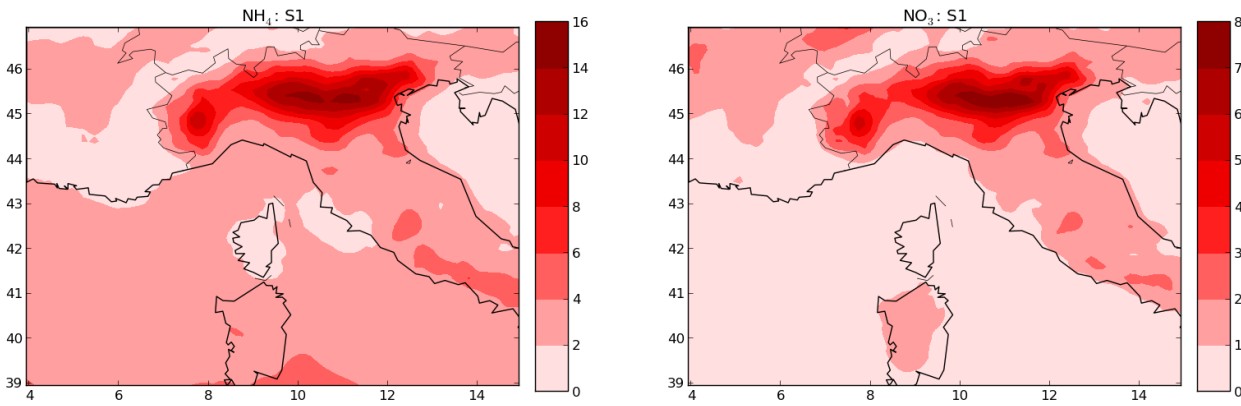

**Figure D5.** Maps of the concentrations of $NH_4$ (left panel) and $NO_3$ (right panel) in $PM_{10}$ during the summer 2013.

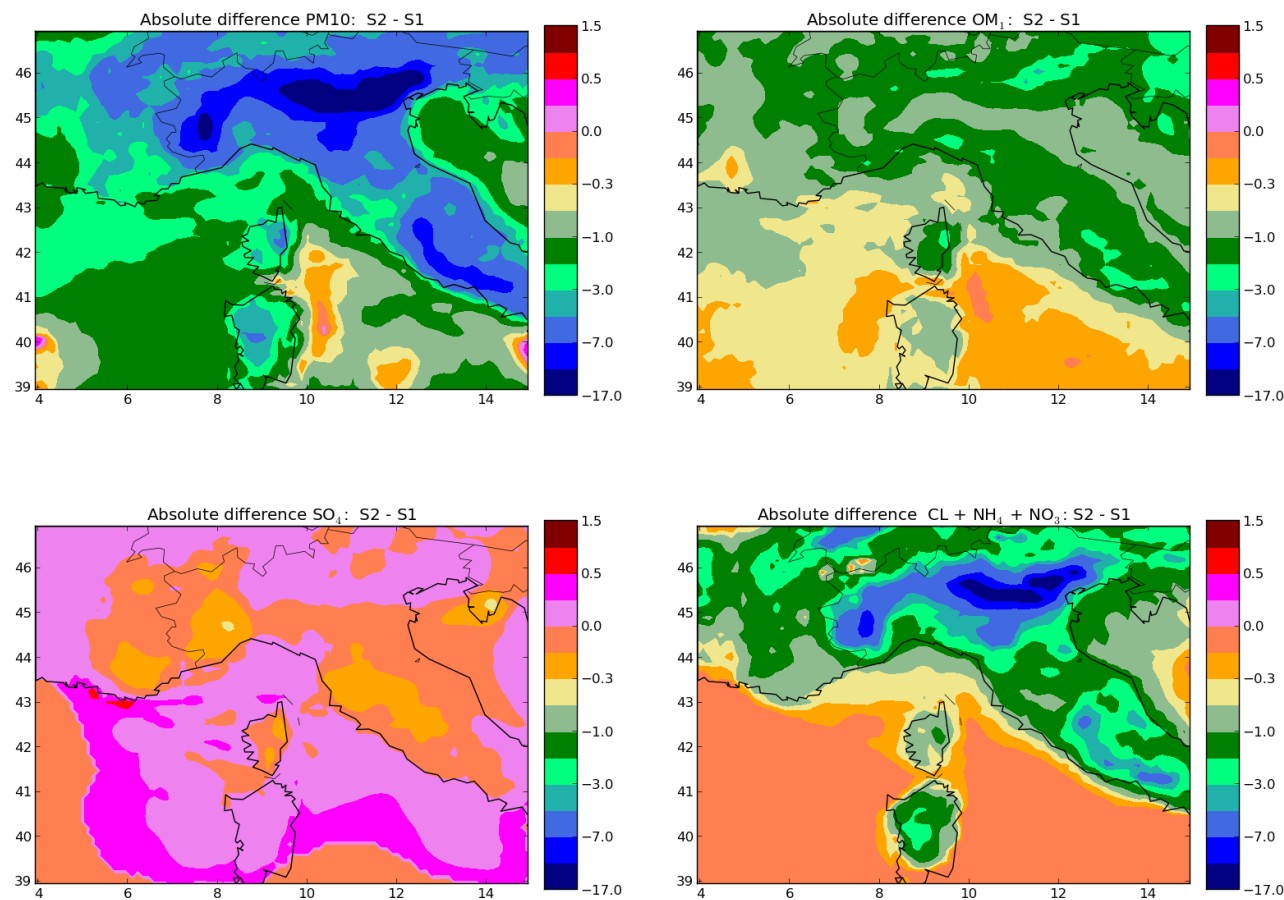

**Figure D6.** Maps of the absolute differences of the concentrations of $PM_{10}$ (upper left panel), $OM_1$ (upper right panel), sulfate (lower left panel) and other inorganics (nitrate, ammonium and chloride) (lower right panel) in % between S1 and S2 (right panel) during the summer 2013.

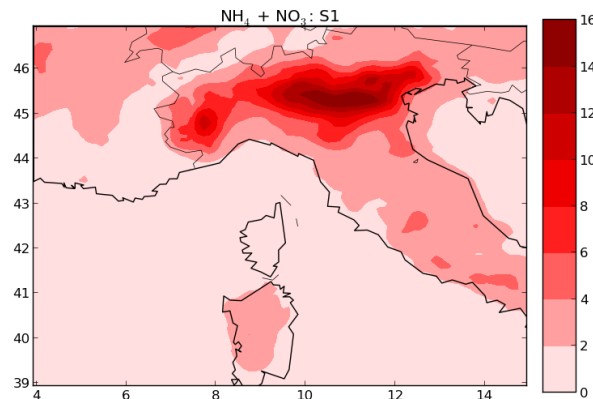

**Figure D7.** Map of the concentrations of $NH_4 + NO_3$ in $PM_{10}$ during the summer 2013.

# Appendix E: Biogenic VOCs

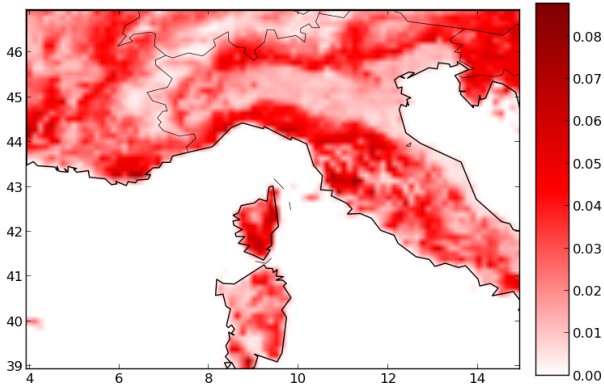

**Figure E1.** Maps of the emission rates of biogenic VOCs (isoprene and terpene) during the summer 2013 in $\mu$g m$^{-2}$.s$^{-1}$.

# Appendix F:  Meteorological evaluation during the flight of 10 July 2014

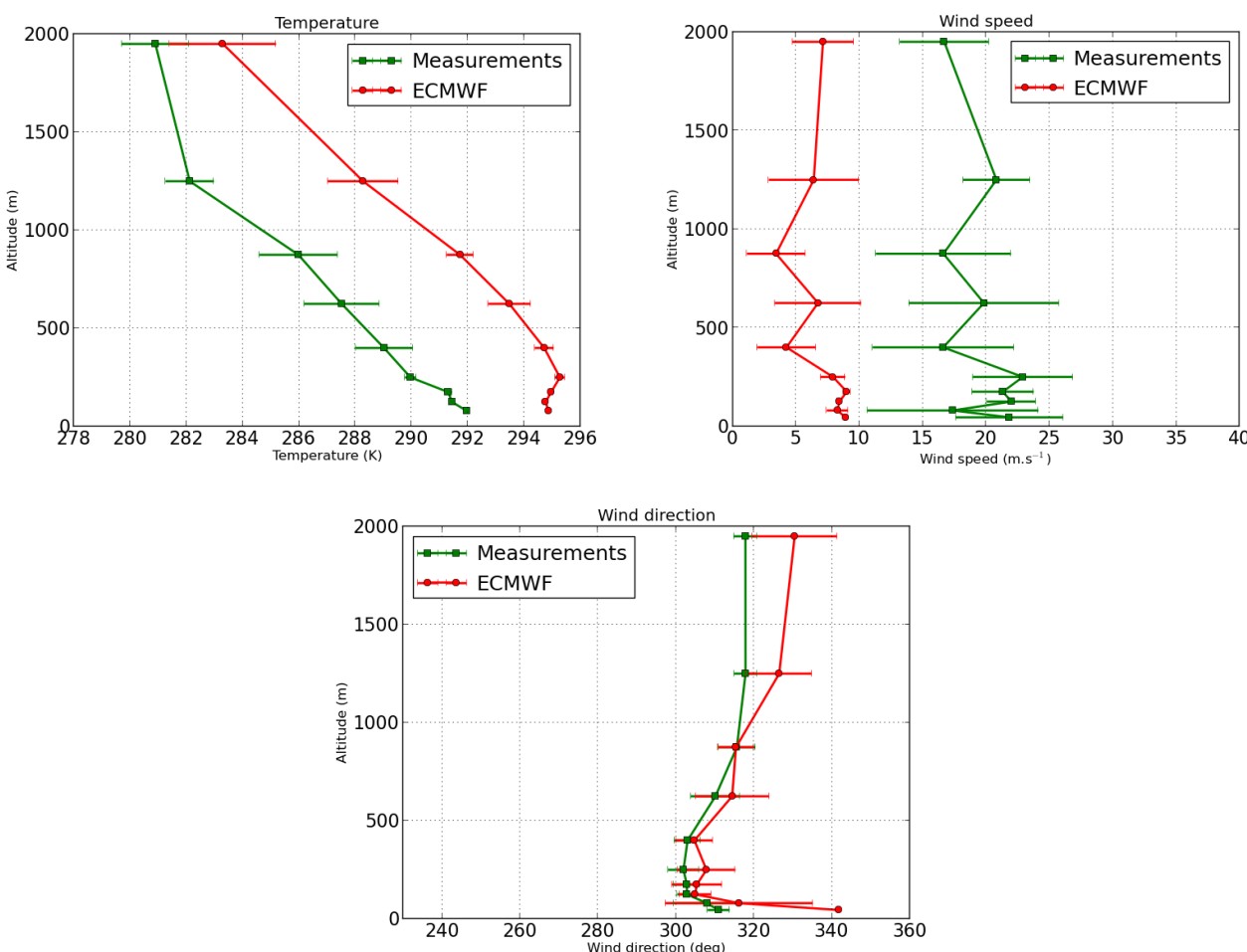

**Figure F1.** Comparison of temperature (upper right panel), wind speed (upper left panel) and wind direction (lower panel) during the flight of 10 July 2014