# Peer review of "Aerosol sources in the western Mediterranean during summertime: A model-based approach"

_Atmospheric Chemistry and Physics, 2017_

## Referee Comment (RC1) · Anonymous Referee #3 · 30 Jan 2018

This study presents the sensitivity of aerosols and their chemical composition over the Eastern Mediterranean as calculated by different model simulations performed in the framework of their ChArMEx experiment. The manuscript is very well organized and easy to follow, with a good level of English language. The manuscript is suitable for publication in ACP provided that the below minor comments are addressed in a revised version.

Material and Methods 1) Is the WRF model configured with 1-way or 2-way nested?

2) It would be good if more background is provided on how these different configurations for meteorology are designed.

3) It is not clear from the text that both EMEP emissions and HTAP emissions are used

in the simulations. Refer to Table 1.

4) Give more information on how the aerosols and organics are calculated in the model. Is it VBS that is used? Is the aerosol module operating on modal or sectional bins?

5) Make it clear that dust emissions are not calculated in the model but only provided from the boundaries. Also provide information on how the boundary conditions are calculated. It is only in the results section that MOZART model is mentioned.

6) Give more details on the SSE calculations. Is the surf zone included, how, or is only the open sea emissions calculated?

7) Table 1 and 2 should be explained in the text and provide the motivation and the reasons for these different scenarios more clearly.

Results

8) Why the time series analyses for 2012 shown in the appendix? This would also show the extend of the dust break contribution to the levels.

9) Explain Table 7 more clearly (in the caption maybe). What are the differences showing, period mean? What is the background of using normalized RMSE to show the sensitivity of the different inputs?

10) As Table 7 shows, majority of the sensitivity simulation target 2013. Therefore, please also show the composition of PM in 2013 too to assist the discussions.

11) Add that these are observed composition in Figure 3 caption.

12) What does the 0.04±0.03 show in the Table 8?

13) Page 15, Line 2. Normalized RMSE varies between 44 to 267%, not 48?

Conclusions

What is the general conclusion of the study?

---

## Referee Comment (RC2) · Anonymous Referee #1 · 1 Feb 2018

This modeling study presents the aerosol results of the air-quality model Polyphemus applied to western Mediterranean in summer periods during the ChArMEx campaigns. In their previous publication (Chrit et al., 2017) the authors focused on organic aerosols simulated for the same periods. In this manuscript, modelled PM10, PM1, OM1 and inorganic aerosols were evaluated against measurements at the same site in Corsica as in the previous publication. In addition, sensitivity of model results to meteorological model, anthropogenic emission inventory and some model parameterization was analyzed. The air pollution over the Mediterranean is a significant topic and this manuscript might contribute to the research in this area. There are however, some critical issues to be clarified, addressed and revised before the manuscript can be considered for publication.

My main concern is about the anthropogenic emissions. According to the manuscript, emission inventory for 2010 was used for simulations of 2012, 2013 and 2014. There is no description in the manuscript how the 2010 emissions were adjusted to other years. This needs clarification and justification. If they were not somehow scaled, then it should be discussed in which emissions and sectors (traffic, ships, industry, etc.) differences between years are expected and how much this would affect the results.

Another weak point in the manuscript is the airborne evaluation (section 4.4). The method for comparison of model results with measurements is poorly described. Meteorological evaluation is missing for this period which seems to have different wind speed and direction than the other periods. The model performance for meteorology is very important for the interpretation of results. The method chosen to compare the model results with airborne measurements in Figs. 4 and 5 does not seem to be suitable and therefore the interpretation of various sensitivity simulations is difficult.

**General comments:**

Although they might be published elsewhere, a brief description about the model and input parameters are needed to be given in this manuscript (in Sect. 2) as well (e.g. chemical mechanism, aerosol model, VBS?, particle size (modal or sectional?), boundary conditions, vertical resolution, model top height, which layer is compared with measurements?) Sensitivity simulations for inorganic aerosols should be described earlier in this section (not first in 4.4).

Please do not use SIA (Secondary Inorganic Aerosols) as sum of Cl, $NO_3$, $NH_4$. SIA usually refers to ammonium nitrate and sulfate ($SO_4$, $NO_3$, $NH_4$).

Please avoid one-sentence paragraphs (e.g. page 9, lines 17-18).

**Specific comments:**

Title: Although the title is "*Aerosol sources …*" there is no clear conclusion about the sources. It is more a sensitivity analysis of the model results.

P2, L4: Please consider citing more recent studies.

P2, L9: Di Biagio et al. (2015)

P2, L19-20: *"Difficulties in modeling aerosol concentrations..",* It is not clear what is meant here; all or only inorganic aerosols? Please revise it.

P4: Was ECMWF data in mother domain ($0.25^o$x$0.25^o$) used also for the nested domain with a finer horizontal resolution? If yes, please describe how they were adapted to the finer resolution.

P5, L12: Please add the version of MEGAN used in the simulations. Which meteorological data was used to calculate biogenic emissions? Were biogenic emissions re-calculated using different meteorological data for the sensitivity simulations? Please clarify it and add to the text.

P5, L13-14: Using emissions of 2010 for simulations of 2012, 2013, 2014 needs some discussion. If 2010 data were used without any scaling for other years, it should be justified and it should also be discussed in which sectors (traffic, ships, industry, etc.) differences between years are expected and how much this would affect the results.

P6, L3-4: *"inorganic aerosols (chloride, nitrate, ammonium)"* Please remove *"inorganic aerosols"*, just write those species relevant for partitioning.

P8, L2: Please make it clear:  results from which model layer were used for comparison with measurements, was any interpolation applied?

P8, L10: '*The inorganic precursors $HNO_3$, HCl and $SO_2$ were measured…"* $NH_3$ is one of the most important precursors for inorganic aerosols. Were there any measurements of ammonia? Some information about ammonia emissions (temporal profiles, uncertainties) would be useful for discussion of SIA.

P 8, L30: *"…Appendix A of comparison…"* please replace "*of*" with "and"

P9, L1: Although the mean values over the whole period look satisfactory, Fig. B1 shows bias as high as about 5K in daily points. Deviation in hourly values might be even higher. Please consider the effect of T bias on especially nitrate discussions.

P9, L6-7: An explanation is needed for why ECMWF performs better in spite of its coarser resolution than WRF.

P9, L17-18: "*Tables 5*": should be "Table 5"

P9, L19-22: Authors might consider using recommended statistical parameters for the discussion in meteorological-model evaluation (EPA 2007). Evaluation of meteorology for summer 2014 is not shown. Since wind direction seems to be different than other summers, it is necessary to know how the model performance is for discussion in Section 4.4.

P11, L22: "*probably because of higher occurrence of transported desert dust in 2012*" Please provide some evidence to support this hypothesis.

P11, L23-24: "*.. PM1 concentration is slightly over-estimated during summer 2013..*" On Table 7 however, PM1 in 2013 seems to be under-estimated. Please check and revise the sentence.

P12, L1-13: In this section, it is not clear whether the discussion is about the measured or modeled composition (also in Fig. 3). Please revise it.

P12, L7: BC appears for the first time here. How was BC measured?

P14, L6-9: Justification of underestimated ammonium (even more with EMEP emissions) based only on ship NOx emissions sounds not completely right. It might also be due to differences in $NH_3$ emissions between the two inventories.

P14, L20-23: Please consider revising the sentence "*For nitrate, the total nitrate (gas + particle phase) is under-estimated between 21 and 26 July 2013 (2.7 µ g m−3 in the measurements and 6.6 µ g m−3 simulated), and most of it is in the gas phase (only 0.4 µ g m−3 in the particle phase in the measurements and 0.2 simulated)*"
If measurements are 2.7 and simulations are 6.6, total nitrate is not underestimated but overestimated. It would also be necessary to show these numbers somewhere.

P15, L2-3: "*By influencing biogenic emissions, meteorology affects the formation of organics and hence the formation of inorganics (because of formation of organic nitrate).*" This sentence needs some justification and references (e.g. for organic nitrate formation Ng et al. 2017, for inorganic nitrate formation Aksoyoglu et al., 2017).
Meteorological parameters affecting biogenic emissions are mainly temperature and radiation. How much can be the difference in biogenic emissions due to using another meteorological model (due to differences in T and radiation) to cause such a significant change in inorganics shown in Table 10? If it is difficult

to provide any data to support the role of meteorology via biogenic emissions, then please add some discussion using other published studies.

P15, L5: *"hence the oxidants and the nitrate formation"* Please make it clear whether it is organic or inorganic nitrate.

P15, L9-18: Evaluation of meteorological parameters for the period of 9-10 July 2014 when airborne measurements were performed is missing. How was the modelled wind speed and direction? Which model layer concentrations were compared to aircraft measurements?

P15, L19-21: Please remove this information from here and give in the following corresponding sections.

P17, L1-2: Can the over estimation of sulfate be due to emissions? How were the emissions from ship stacks treated? If they were vertically distributed, to which model layer were they emitted?

P18, L2: "*ammonium and nitrate and chloride..*" please replace the first "and" with a comma.

P18, L28-29: Please do not use SIA for chloride, nitrate and ammonium.

P18, L30- 31: *"OM1 concentrations are high nearby locations of high biogenic emissions"* Is there any evidence (high biogenic SOA) to support this sentence or any reference for high biogenic emissions in those regions?

P19, L9: This sentence suggests that biogenic emissions were recalculated using different meteorology so that impacts on biogenic emissions were taken into account. If it is the case, this should be described in the methods section.

P20, L5-8: It needs some discussion how sulfate (and other inorganic aerosols) is affected by sea-salt parameterization.

P20, L31: In addition to temperature, other parameters such as radiation (for photolysis) and humidity also affect the reactions and formation of secondary products.

Table 10: Measured and modeled means of nitrate are similar (slight overestimation) while MFB is negative -24. Is this correct?

**Figures:**
Fig. 1: Is it reasonable to have such a low PBLH at noon over the sea north of Corsica? Can it be validated?

Fig. 3: Are these measured or modeled compositions? Please revise the caption.

Figs. 4-5: This is the first place where "model levels" are mentioned. This has to be described in the methods section.

These figures are difficult to understand. Averaged vertical profiles do not give any information about the location. Looking at the purple lines in Fig 1, it seems to me that the location can be either close to the southern coast of France, over the sea in the middle between French coast and Corsica or close to the Corsican coast. In addition, profiles seem to have a wide range (lines at certain altitudes) which makes the average values very uncertain. It would be more useful to extract model data from the corresponding grid cells and layers along the flight path for comparison with measurements. Because of these uncertainties, discussion in whole section 4.4 about sensitivity simulations does not make much sense.

Fig. B1: WRF Lambert OBSGRID is worst (bias 5K) in spite of nudging, why? Please replace "wind module" with "wind speed"

Fig. B2: What is the reason of less variation in 2013 with respect to 2012?

Fig. C1. S1 cannot be seen in the OM1 figure. Are S1 and S5 overlapped?
Scales in x-axes are different, i.e. PM10 July06-July31, PM1 June06-Aug05, OM1 Jun07-Aug05. It would be better if they were consistent, also with the meteorological figures (B1)

Fig. C2: Measured PM10 $SO_4$ looks sometimes lower than measured PM1 $SO_4$ (eg. June 15). Please check and/or justify.

Fig. C3: The modeled PM10 and PM1 $NO_3$ look exactly the same in all simulations, suggesting that coarse nitrate was not modeled. If this is the case, it should be mentioned in the methods section. Then there is no need to show the figure for PM10 $NO_3$ (same for $NH_4$ too).

Fig. D1: Please give the size fraction for the figures in the lower 2 panels.
Map for $Cl+NO_3+NH_4$ (please do not call it SIA) shows high concentrations (up to 24 microg/m$^3$). Please give some information which species is contributing more to this high level. Considering highly polluted Po Basin and high ammonia emissions in the region, high concentrations are very likely due to ammonium nitrate. It would make more sense to show maps of $NO_3$ and $NH_4$ separately and not to sum with Cl.

Fig. D2. It would be useful to see also the absolute differences.

Fig. D3: please separate Cl from $NO_3$ and $NH_4$. It is probably dominated by Cl.

Please revise the following references: Chrit et al 2017, Cholakian et al. 2017

**References**

EPA, U.: Guidance on the use of models and other analyses for demonstrating attainment of air quality goals for ozone,  PM2. 5, and regional haze, US Environmental Protection Agency, Office of Air Quality Planning and Standards, 2007.

Aksoyoglu, S., Ciarelli, G., El-Haddad, I., Baltensperger, U., and Prévôt, A. S. H.: Secondary inorganic aerosols in Europe: sources and the significant influence of biogenic VOC emissions, especially on ammonium nitrate, Atmos. Chem. Phys., 17, 7757-7773, https://doi.org/10.5194/acp-17-7757-2017, 2017.

Ng, N. L., Brown, S. S., Archibald, A. T., Atlas, E., Cohen, R. C., Crowley, J. N., Day, D. A., Donahue, N. M., Fry, J. L., Fuchs, H., Griffin, R. J., Guzman, M. I., Herrmann, H., Hodzic, A., Iinuma, Y., Jimenez, J. L., Kiendler-Scharr, A., Lee, B. H., Luecken, D. J., Mao, J., McLaren, R., Mutzel, A., Osthoff, H. D., Ouyang, B., Picquet-Varrault, B., Platt, U., Pye, H. O. T., Rudich, Y., Schwantes, R. H., Shiraiwa, M., Stutz, J., Thornton, J. A., Tilgner, A., Williams, B. J., and Zaveri, R. A.: Nitrate radicals and biogenic volatile organic compounds: oxidation, mechanisms, and organic aerosol, Atmos. Chem. Phys., 17, 2103-2162, https://doi.org/10.5194/acp-17-2103-2017, 2017.

---

## Author Comment (AC1) · 13 Jun 2018

This modeling study presents the aerosol results of the air-quality model Polyphemus applied to western Mediterranean in summer periods during the ChArMEx campaigns. In their previous publication (Chrit et al., 2017) the authors focused on organic aerosols simulated for the same periods. In this manuscript, modelled PM10, PM1, OM1 and inorganic aerosols were evaluated against measurements at the same site in Corsica as in the previous publication. In addition, sensitivity of model results to meteorological model, anthropogenic emission inventory and some model parameterization was analyzed. The air pollution over the Mediterranean is a significant topic and this manuscript might contribute to the research in this area. There are however, some critical issues to be clarified, addressed and revised before the manuscript can be considered for publication.

My main concern is about the anthropogenic emissions. According to the manuscript, emission inventory for 2010 was used for simulations of 2012, 2013 and 2014. There is no description in the manuscript how the 2010 emissions were adjusted to other years. This needs clarification and justification. If they were not somehow scaled, then it should be discussed in which emissions and sectors (traffic, ships, industry, etc.) differences between years are expected and how much this would affect the results.

Concerning anthropogenic emissions, EMEP gridded emissions are released every year. However, countries only report gridded emissions every 4 to 5 years (2010, 2015,..), and emissions for 2015 were only release recently, after this work was finished. Gridded emissions of 2012 and 2013 are therefore estimations, and we preferred to use the reported emissions for 2010, especially as the EDGAR-HTAP inventory is also only available in 2005 and 2010.
According to the official non-gridded emissions (http://www.ceip.at/ms/ceip_home1/ceip_home/data_viewers/official_tableau/), NOx and VOC emissions have decreased by 11% over France between 2010 and 2013. Although this 11% may appear high in terms of emissions, its impact on aerosol concentrations is lower, probably a few percents. Furthermore, this decrease is subject to a high level of uncertainties. A large fraction of NOx emissions is due to traffic, and traffic NOx emissions may be under-estimated according to Ntziachristos et al. (2016). Concerning PM emissions, they have also decreased by 8% over France between 2010 and 2013. This is not enough to affect consequently our results, since most of the particle compounds described in this article are secondary. The gridded estimated ship emissions for 2010 and 2013 are similar.

Another weak point in the manuscript is the airborne evaluation (section 4.4). The method for comparison of model results with measurements is poorly described. Meteorological

evaluation is missing for this period which seems to have different wind speed and direction than the other periods. The model performance for meteorology is very important for the interpretation of results. The method chosen to compare the model results with airborne measurements in Figs. 4 and 5 does not seem to be suitable and therefore the interpretation of various sensitivity simulations is difficult.

The method for comparison of model results with measurements is now better described, and a meteorological evaluation is added in Appendix F.

With this airborne to measurement comparison, we aim to assess the concentrations above the Mediterranean sea. Therefore, we averaged vertical profiles not at all locations, but only at locations above the sea, where the flight flew at low altitudes and where the boundary layer was high enough. This approach is similar to the one suggested by the reviewer: we extracted model data from the corresponding grid cells and layers along the flight path to do so, and we averaged only data for cells with similar characteristics (over the sea, high PBL height and low flight altitude).

For clarity, the sentences

"For the comparisons of inorganic concentrations to airborne measurements, the reference simulation S1 is run a few days during the summer 2014 and it is compared to the observed concentrations when the flight is below 800 m.a.s.l. and where the boundary layer is spatially uniform (above 1200 m). The transects where model to measurements are performed are indicated by purple crosses/lines in Figure 1."

Are replaced by

"For the comparisons of inorganic concentrations to airborne measurements, the reference simulation S1 is run a few days during the summer 2014. The simulated concentrations are extracted along the flight path from the corresponding grid cells and layers. For the model to measurement comparisons, only the cells were the plane was flying above the sea, at low altitudes (below 800 m.a.s.l.) with a spatially uniform boundary layer (above 1200 m) are considered."

**General comments:**

Although they might be published elsewhere, a brief description about the model and input parameters are needed to be given in this manuscript (in Sect. 2) as well (e.g. chemical mechanism, aerosol model, VBS?, particle size (modal or sectional?), boundary conditions, vertical resolution, model top height, which layer is compared with measurements?) Sensitivity simulations for inorganic aerosols should be described earlier in this section (not first in 4.4).

A brief description of the model and input parameters are added in the revised paper: "Vertically, 24 vertical levels are used in WRF and 14 levels are used in Polair3d/Polyphemus. The heights of the cell interfaces are 0, 30, 60, 100, 150, 200, 300, 500, 750, 1000, 1500, 2400, 3500, 6000 and 12 000 m.."

"The numerical algorithms used for transport and the parameterisations used for dry and wet depositions are detailed in Sartelet et al. (2007). Gas-phase chemistry is modelled with the carbon bond 05 mechanism (CB05) (Yarwood et al., 2005), to which reactions are added to model the formation of secondary organic aerosols (Kim et al., 2011; Chrit et al. 2017). The SIze REsolved Aerosol Model (SIREAM; Debry et al., 2007) is used for simulating the dynamics of the aerosol size distribution by coagulation and condensation/evaporation. SIREAM uses a sectional approach and the aerosol distribution is described here using 20 sections of bound diameters: 0.01, 0.0141, 0.0199, 0.0281, 0.0398, 0.0562, 0.0794, 0.1121, 0.1585, 0.2512, 0.3981, 0.6310, 1.0, 1.2589, 1.5849, 1.9953, 2.5119, 3.5481, 5.0119, 7.0795 and 10.0 µm. The condensation/evaporation of inorganic aerosols is determined using the thermodynamic model ISORROPIA (Nenes et al., 1998) with a bulk equilibrium approach in order to compute the partitioning between the gaseous and particle phases of aerosols. Because the concentrations and the partitioning between gaseous and particle phases of chloride, nitrate, ammonium is strongly affected by condensation/evaporation and reactions with other pollutants, sensitivities of these concentrations to hypothesis used in the modeling (thermodynamic equilibrium, mixed sea-salt and anthropogenic aerosols) are also performed (section 4.4.2).

For organic aerosols, the gas–particle partitioning of the surrogates is computed using SOAP assuming bulk equilibrium (Couvidat and Sartelet, 2015), and bulk equilibrium is also assumed for SOA partitioning. The gas–particle partitioning of hydrophobic surrogates is modeled following Pankow (1994), with absorption by the organic phase (hydrophobic surrogates). The gas– particle partitioning of hydrophilic surrogates is computed using the Henry's law modified to extrapolate infinite dilution conditions to all conditions using an aqueous-phase partitioning coefficient, with absorption by the aqueous phase (hydrophilic organics, inorganics and water). Activity coefficients are computed with the thermodynamic model UNIFAC (UNIversal Functional group; Fredenslund et al., 1975). After condensation/evaporation, the moving diameter algorithm is used for mass redistribution among size bins. As detailed in Chrit et al. (2017), anthropogenic intermediate/semi-volatile organic compounds (I/S-VOC) emissions are emitted as three primary surrogates of different volatilities (characterized by their saturation concentrations $C*$: $\log(C*) = -0.04, 1.93, 3.5$). The ageing of each primary surrogate is represented through a single oxidation step, without NOx dependence, to produce a secondary surrogate of lower volatility ($\log(C*) = -2.4, -0.064, 1.5$ respectively) but higher molecular weight. Gaseous I/S-VOC emissions are missing from emission inventories, they are estimated here as detailed in Zhu et al. (2016) by multiplying the primary organic emissions (POA) by 1.5, and by assigning them to species of different volatilities. A sensitivity study where I-S/VOC emissions are not taken into account is also performed."

"The boundary conditions for the European simulation are calculated from the global model MOZART4 (Horowitz et al., 2003) (https://www.acom.ucar.edu/wrf-chem/mozart.shtml), and those for the Mediterranean domain are obtained from the European simulation. Mineral dust emissions are not calculated in the model, but are provided from the boundaries, and their heterogeneous reactions to form nitrate and sulfate are not taken into account."

For clarity about the description of the sensitivity simulations, Section 2.1 is renamed "Simulation set-up and alternative parameterizations", and the following sentences are added:
 "In order to simulate aerosol formation over the western Mediterranean, the Polair3d/Polyphemus air quality model is used, with the set-up described in Chrit et al. (2017) and summarized here. For

parameters/parameterizations that are particularly attached to uncertainties (anthropogenic emissions, meteorology, sea-salt emissions, modeling of condensation/evaporation), the alternative parameters/ parameterizations that are used in the sensitivity studies are also detailed for emissions and meteorology. For computational reasons, alternative parameterizations for the modeling of condensation/evaporation are only used in the comparisons to airborne measurements in section 4.4, where they are detailed. "

Please do not use SIA (Secondary Inorganic Aerosols) as sum of Cl, $NO_3$, $NH_4$.
SIA usually refers to ammonium nitrate and sulfate ($SO_4$, $NO_3$, $NH_4$).
SIA is replaced in the revised paper by chloride, nitrate and ammonium, or by the term VIA (Volatile Inorganic Aerosols)

Please avoid one-sentence paragraphs (e.g. page 9, lines 17-18).

In the revised paper, this sentence is moved to the beginning of the paragraph below this one.

**Specific comments:**

Title: Although the title is "*Aerosol sources …*" there is no clear conclusion about the sources. It is more a sensitivity analysis of the model results.

The conclusion was rewritten to better describe the aerosol sources.

- Sulfate originates mostly from maritime traffic. Furthermore, maritime traffic leads to the formation of oxidants that in turn enhance the formation of biogenic aerosols, with the potential formation of organic nitrate and organo sulfate.
- Organics are mostly from a biogenic origins. Even if the contribution of sea-salt emissions to organic concentrations is low, organic concentrations are strongly influenced by sea-salt emissions, because they partition between the gas and the particle phases and they are hydrophilic. This underlines the need to better characterize the properties (affinity with water) of secondary organic aerosols.
- Secondary pollutants, such as nitrate, ammonium and chloride, as the particle-phase concentrations are strongly influenced by the gas/particle phase partitioning, because a high percentage of their concentration is in the gas phase. This underlines the need to develop aerosol models able to represent accurately this gas-phase partitioning.
- There is a high sensitivity of secondary pollutants (inorganics and organics) to meteorology, stressing the importance of accurate meteorological modeling and the potential strong influence of climate change on the concentrations of these secondary pollutants.

P2, L4: Please consider citing more recent studies.

Debevec et al. 2017, Doche et al., 2014, Menut et al., 2015, Nabat et al., 2013 and Safieddine et al., 2014 are added as references in the revised paper.

P2, L9: Di Biagio et al. (2015)
This reference was corrected in the revised paper.

P2, L19-20: *"Difficulties in modeling aerosol concentrations.."*, It is not clear what is meant here; all or only inorganic aerosols? Please revise it.
It is meant "all". This sentence was replaced in the revised paper by " …. Difficulties in modeling atmospheric particles, …"

P4: Was ECMWF data in mother domain ($0.25^{o}$x$0.25^{o}$) used also for the nested domain with a finer horizontal resolution? If yes, please describe how they were adapted to the finer resolution.

In the reference simulation, ECMWF was used for both domains. ECMWF data were interpolated to the model grid. For clarity, the sentence "In the reference simulation, meteorological data are provided by the European Center for Medium-RangeWeather Forecasts (ECMWF) model (horizontal resolution: 0.25°×0.25°)." is replaced by "In the reference simulation, meteorological data are provided by the European Center for

Medium-RangeWeather Forecasts (ECMWF) model (horizontal resolution: 0.25°×0.25°), which are interpolated to the Europe and Mediterranean domains. ”

P5, L12: Please add the version of MEGAN used in the simulations. Which meteorological data was used to calculate biogenic emissions? Were biogenic emissions re-calculated using different meteorological data for the sensitivity simulations? Please clarify it and add to the text.

The model uses the standard MEGAN LAIv database (MEGAN-L, Guenther et al. 2006) with the EFv2.1 dataset. Yes, the biogenic emissions were re-calculated using the different meteorological data for the sensitivity simulations.

The sentence "Biogenic emissions are estimated using Model of Emissions of Gases and Aerosols from Nature (MEGAN) (Guenther et al., 2006)." was modified in the revised paper: "Biogenic emissions are estimated using Model of Emissions of Gases and Aerosols from Nature (MEGAN) with the standard MEGAN LAIv database (MEGAN-L, Guenther et al., 2006) and the EFv2.1 dataset. For the different simulations, these emissions are recalculated with the meteorological data used for transport."

P5, L13-14: Using emissions of 2010 for simulations of 2012, 2013, 2014 needs some discussion. If 2010 data were used without any scaling for other years, it should be justified and it should also be discussed in which sectors (traffic, ships, industry, etc.) differences between years are expected and how much this would affect the results.

Concerning anthropogenic emissions, EMEP gridded emissions are released every year. However, countries only report gridded emissions every 4 to 5 years (2010, 2015,..), and emissions for 2015 were only release recently, after this work was finished. Gridded emissions of 2012 and 2013 are therefore estimations, and we preferred to use the reported emissions for 2010, especially as the EDGAR-HTAP inventory is also only available in 2005 and 2010.

According to the official non-gridded emissions (http://www.ceip.at/ms/ceip_home1/ceip_home/data_viewers/official_tableau/), NOx and VOC emissions have decreased by 11% over France between 2010 and 2013. Although this 11% may appear high in terms of emissions, its impact on aerosol concentrations is lower, probably a few percents. Furthermore, this decrease is subject to a high level of uncertainties. A large fraction of NOx emissions is due to traffic, and traffic NOx emissions may be under-estimated according to Ntziachristos et al. (2016). Concerning PM emissions, they have also decreased by 8% over France between 2010 and 2013. This is not enough to affect consequently our results, since most of the particle compounds described in this article are secondary.

P6, L3-4: *"inorganic aerosols (chloride, nitrate, ammonium)"* Please remove *"inorganic aerosols"*, just write those species relevant for partitioning.

"Inorganic aerosols" is removed from this sentence in the revised paper.

P8, L2: Please make it clear: results from which model layer were used for comparison with measurements, was any interpolation applied?

No interpolation was applied. For clarity, the sentence: "Simulated concentrations are compared to ground-based measurements performed at Ersa" is replaced by "Simulated concentrations in the first vertical level of the model are compared to ground-based measurements performed at Ersa"

P8, L10: '*The inorganic precursors HNO₃, HCl and SO₂ were measured…"* NH$_3$ is one of the most important precursors for inorganic aerosols. Were there any measurements of ammonia? Some information about ammonia emissions (temporal profiles, uncertainties) would be useful for discussion of SIA.

Unfortunately, due to instrumentation problem during the measurement campaign, the measurements of ammonia are not available. Uncertainties on the spatial distribution of ammonia emissions are partly taken into account in the differences between EDGAR-HTAP and EMEP emission inventories. We did not explore the effect of uncertainties in the temporal profiles, although we expect them to be lower than the uncertainties on gas/phase partitioning. Because, over the Mediterranean in summer, ammonia is expected to be mostly in the gas phase, the particle concentration is mostly dependent on the modeled gas/particle partitioning.

P 8, L30: *"…Appendix A of comparison…"* please replace "*of*" with "and"
"of" was replaced by "and" in the revised paper.

P9, L1: Although the mean values over the whole period look satisfactory, Fig. B1 shows bias as high as about 5K in daily points. Deviation in hourly values might be even higher. Please consider the effect of T bias on especially nitrate discussions.

The deviation is now taken into account in the statistics MB, which is added to the paper. A discussion of the effect of T bias on nitrate formation is added to section 4.3.1.

P9, L6-7: An explanation is needed for why ECMWF performs better in spite of its coarser resolution than WRF.
It is added to the paper that WRF was forced here with NCEP meteorological fields for initial and boundary conditions. NCEP has a lower resolution than WRF (1º×1º grid spacing). On small computational domains, such as the ones used here, the performance of WRF is strongly linked to the performance of the meteorological model used for the forcing.

P9, L17-18: "*Tables 5*": should be "Table 5"
"Tables 5" is replaced by "Tables 3, 4, 5, 6" in the revised paper.

P9, L19-22: Authors might consider using recommended statistical parameters for the discussion in meteorological-model evaluation (EPA 2007). Evaluation of meteorology for summer 2014 is not shown. Since wind direction seems to be different than other summers, it is necessary to know how the model performance is for discussion in Section 4.4.
A comparison with Emery et al. 2001 performance criteria is added in the revised paper:
"As mentioned in EPA 2007 report, Emery et al. (2001) proposed benchmarks for temperature (mean bias (MB) within ±0.5 K and gross error (GE) < 2.0 K), wind speed (MB within ±0.5 m s−1 and RMSE < 2ms−1 ) and wind direction (MB within ±10° and GE < 30°). McNally (2009) suggested an alternative set of benchmarks for temperature (MB within ±1.0 K and GE < 3.0 K)."
The statistics are then discussed through the section on the model to measurements comparisons. The comparison of the modeled and measured meteorological data during the flight of 10 July 2014 is shown in Appendix F.

P11, L22: *"probably because of higher occurrence of transported desert dust in 2012"*
Please provide some evidence to support this hypothesis.

Nabat et al. 2015 showed during the ChArMEx/TRAQA (TRansport and Air QuAlity) campaign that focused on the characterization of the polluted air masses over the Mediterranean basin during the summer 2012 that a particularly intense dust event has been measured with different observation means (balloons, aircraft, surface and remote-sensing measurements).

This reference is added in the revised paper: "…probably because of higher occurrence of transported desert dust in 2012 (Nabat et al. 2015)"

P11, L23-24: *".. PM1 concentration is slightly over-estimated during summer 2013."* In Table 7 however, PM1 in 2013 seems to be under-estimated. Please check and revise the sentence.

The sentence *".. the mean $PM_1$ concentration is slightly overestimated during summer 2013.."* is replaced in the revised paper by: " … the mean $PM_1$ concentration is slightly under-estimated during summer 2013 …"

P12, L1-13: In this section, it is not clear whether the discussion is about the measured or modeled composition (also in Fig. 3). Please revise it.

The composition discussed in Fig 3 and in this section are the simulated compositions. It is added in Fig 3 caption and in the text of the revised paper: "… above. Figure 3 shows the simulated composition of …" and "… variability. According to simulation, inorganic aerosols account for a large part of the $PM_{10}$ mass: during the summer campaign periods of 2012 and 2013…"

P12, L7: BC appears for the first time here. How was BC measured?

The percentage mentioned here is the simulated percentage of BC. It is now easier to understand in the revised version, as "According to simulation" was added at the beginning of the description of the composition of $PM_{10}$.

P14, L6-9: Justification of underestimated ammonium (even more with EMEP emissions) based only on ship NOx emissions sounds not completely right. It might also be due to differences in $NH_3$ emissions between the two inventories.

The influence of the differences of NH3 emissions between the two inventories are very low over the sea (lower than 1%), because NH3 shipping emissions are very low over the Mediterranean Sea. Therefore, we believe that the differences in NH3 emissions between the inventories are limited over the sea.

P14, L20-23: Please consider revising the sentence *"For nitrate, the total nitrate (gas + particle phase) is under-estimated between 21 and 26 July 2013 (2.7 µ g m−3 in the measurements and 6.6 µ g m−3 simulated), and most of it is in the gas phase (only 0.4 µ g m−3 in the particle phase in the measurements and 0.2 simulated)"*
If measurements are 2.7 and simulations are 6.6, total nitrate is not underestimated but overestimated. It would also be necessary to show these numbers somewhere.

Yes, indeed, this "underestimated" is replaced by "overestimated". We did not show these numbers in the Tables because measurements were only performs for a few days, and not over the whole measurement periods.

P15, L2-3: *"By influencing biogenic emissions, meteorology affects the formation of organics and hence the formation of inorganics (because of formation of organic nitrate)."* This sentence needs some justification and references (e.g. for organic nitrate formation Ng et al. 2017, for inorganic nitrate formation Aksoyoglu et al., 2017). Meteorological parameters affecting biogenic emissions are mainly temperature and radiation. How much can be the difference in biogenic emissions due to using another meteorological model (due to differences in T and radiation) to cause such a significant change in inorganics shown in Table 10? If it is difficult to provide any data to support the role of meteorology via biogenic emissions, then please add some discussion using other published studies.

References and discussions are added to the revised paper: "By influencing biogenic emissions, meteorology affects the formation of organics (Sartelet et al. 2012), because they are mostly of biogenic origins in summer (Chrit et al. 2017). The influence of meteorology on biogenic emissions also affects the formation of inorganics because of the modification of oxidant concentrations (Aksoyoglu et al. 2017) and because of the formation of organic nitrate (Ng et al., 2017)."

P15, L5: *"hence the oxidants and the nitrate formation"* Please make it clear whether it is organic or inorganic nitrate.
This sentence is modified in the revised paper: "hence the oxidants and both organic and inorganic nitrate formation".

P15, L9-18: Evaluation of meteorological parameters for the period of 9-10 July 2014 when airborne measurements were performed is missing. How was the modelled wind speed and direction? Which model layer concentrations were compared to aircraft measurements?
For clarity about the model layers compared, the sentence "The meteorological fields during this flight are compared with measured data in Appendix F. For the comparisons of inorganic concentrations to airborne measurements, the reference simulation S1 is run a few days during the summer 2014 and it is compared to the observed concentrations when the flight is below 800 m.a.s.l." is replaced by "For the comparisons of inorganic concentrations to airborne measurements, the reference simulation S1 is run a few days during the summer 2014 and the vertical distribution of concentrations is compared to the observed concentrations when the flight is below 800 m.a.s.l."
The comparison of temperature, wind speed and direction is added in Appendix F of the revised paper.

P15, L19-21: Please remove this information from here and give in the following corresponding sections.
The sentences are moved. At the beginning of the section 4.4.1 of the revised paper, we added: "Figure 4 shows the comparison of sulfate to the airborne measurements using different model configurations. Sulfate is the inorganic compound with the highest $PM_1$ concentrations (about 0.54 µg m$^{-3}$)" and we added at the beginning of the section 4.4.2: "Figure 5 shows the comparison of nitrate and ammonium concentrations in $PM_1$. The simulated means of ammonium and nitrate are about 0.32 µg m$^{-3}$ and about 0.14 µg m$^{-3}$ respectively."

P17, L1-2: Can the over estimation of sulfate be due to emissions? How were the emissions from ship stacks treated? If they were vertically distributed, to which model layer were they emitted?

Yes, ship stacks emissions were not vertically distributed but added in the first model level.
 The overestimation of sulfate emissions may be due to the overestimation of sulfate emissions, or to the treatment of ship emissions. This remark is added in the revised paper.
The sentence "This is indicative of the overestimation of sulfate or sulfuric acid emissions, or to the treatment in the model of emissions from ship stacks." is added after the sentence "A comparison of $PM_{10}$ sulfate concentrations for the two simulations show that this is also the case for $PM_{10}$ sulfate concentrations."

P18, L2: "*ammonium and nitrate and chloride..*" please replace the first "and" with a comma.

The first "and" is replaced by a comma in the revised paper.

P18, L28-29: Please do not use SIA for chloride, nitrate and ammonium.

The term SIA is removed from the whole revised paper, and it is replaced by chloride, nitrate and ammonium.

P18, L30- 31: *"OM1 concentrations are high nearby locations of high biogenic emissions"*
Is there any evidence (high biogenic SOA) to support this sentence or any reference for high biogenic emissions in those regions?

A map of biogenic VOCs (terpenes + isoprene) emissions is added in Appendix E of the revised paper, and a reference to this Figure is added in the text of the revised paper.

P19, L9: This sentence suggests that biogenic emissions were recalculated using different meteorology so that impacts on biogenic emissions were taken into account. If it is the case, this should be described in the methods section.

This precision is added in the methods section: "Biogenic emissions are estimated using Model of Emissions of Gases and Aerosols from Nature (MEGAN) with the standard MEGAN LAIv database (MEGAN-L, Guenther et al., 2006) and the EFv2.1 dataset. For the different simulations, these emissions are recalculated with the meteorological data used for transport."

P20, L5-8: It needs some discussion how sulfate (and other inorganic aerosols) is affected by sea-salt parameterization.

The marine sulfate is directly emitted by sea salt (sulfate is assumed to be 4% of the sea-salt emissions as detailed in the section describing the model). The sentence "Although sulfate is little influenced by sea-salt emissions at Ersa (the relative concentration difference is between 0% and 20%), the effect is stronger over the western part of the Mediterranean domain (with relative concentration differences between S3 and S1 between 20% and 60%), where SIA concentrations are also strongly influenced by sea-salt emissions. " is replaced by "As sulfate is assumed to make only 4% of sea-salt emissions (section 2.1), the influence of sea-salt emissions on sulfate concentrations at ERSA is low (the relative concentration difference is between 0% and 20%). The effect is stronger over the western part of the Mediterranean domain (with relative concentration differences between S3 and S1 between 20% and 60%), where sea-salt emissions are stronger. Chloride concentrations are also strongly influenced by sea-salt emissions, as it is directly emitted (it is assumed to make 25% of sea-salt emissions). Nitrate and ammonium concentrations are also

strongly influenced by sea-salt emissions, because of thermodynamic exchanges between the gas and particle phases of chloride, nitrate and ammonium."

P20, L31: In addition to temperature, other parameters such as radiation (for photolysis) and humidity also affect the reactions and formation of secondary products.
In the revised paper, we added: "… the impact on temperature, humidity and radiation, influencing the secondary …"

Table 10: Measured and modeled means of nitrate are similar (slight overestimation) while MFB is negative -24. Is this correct?
The simulated mean is slightly higher than the observed mean. This is due to the high peaks. However, the concentrations of $NO_3$ are overall underestimated. That is why the bias is negative bias even though the modeled mean is slightly higher than the observed mean.

**Figures:**
Fig. 1: Is it reasonable to have such a low PBLH at noon over the sea north of Corsica? Can it be validated?
As shown by Von Engeln et al. 2013, over the Mediterranean Sea, the summer season shows low PBL values (lower than 500 m) possibly due to less contrast between air and sea temperatures [this pattern is also found in Seidel et al. (2012)].

Fig. 3: Are these measured or modeled compositions? Please revise the caption.
These are modeled compositions. It is added in the caption of Fig 3 in the revised paper.

Figs. 4-5: This is the first place where "model levels" are mentioned. This has to be described in the methods section.
In section 2.1 of the revised paper, we added a description of the vertical levels: "The heights of the cell interfaces are 0, 30, 60, 100, 150, 200, 300, 500, 750, 1000, 1500, 2400, 3500, 6000 and 12 000 m."

These figures are difficult to understand. Averaged vertical profiles do not give any information about the location. Looking at the purple lines in Fig 1, it seems to me that the location can be either close to the southern coast of France, over the sea in the middle between French coast and Corsica or close to the Corsican coast. In addition, profiles seem to have a wide range (lines at certain altitudes) which makes the average values very uncertain. It would be more useful to extract model data from the corresponding grid cells and layers along the flight path for comparison with measurements. Because of these uncertainties, discussion in whole section 4.4 about sensitivity simulations does not make much sense.

With this airborne to measurement comparison, we aim to assess the concentrations above the Mediterranean sea. Therefore, we averaged vertical profiles not at all locations, but only at locations above the sea, where the flight flew at low altitudes and where the boundary layer was high enough. This approach is similar to the one suggested by the reviewer: we extracted model data from the corresponding grid cells and layers along the flight path to do so, and we averaged only data for cells with similar characteristics (over the sea, high PBL height and low flight altitude).

For clarity, the sentences

"For the comparisons of inorganic concentrations to airborne measurements, the reference simulation S1 is run a few days during the summer 2014 and it is compared to the observed concentrations when the flight is below 800 m.a.s.l. and where the boundary layer is spatially uniform (above 1200 m). The transects where model to measurements are performed are indicated by purple crosses/lines in Figure 1."

Are replaced by

"For the comparisons of inorganic concentrations to airborne measurements, the reference simulation S1 is run a few days during the summer 2014. The simulated concentrations are extracted from the corresponding grid cells and layers along the flight path. For the model to measurement comparisons, only the cells were the plane was flying above the sea, at low altitudes (below 800 m.a.s.l.) with a spatially uniform boundary layer (above 1200 m) are considered."

Fig. B1: WRF Lambert OBSGRID is worst (bias 5K) in spite of nudging, why? Please replace "wind module" with "wind speed"

We did not assimilate WRF Lambert data with the observations at Ersa but using the NCEP global observations which are quite far from Ersa. That is why the effect of nudging may not provide improved model to measurement comparisons at ERSA. In the revised paper, "wind module" is replace by "wind speed".

Fig. B2: What is the reason of less variation in 2013 with respect to 2012?

The variations between the different models are similar between the 2 years. It looks like there is less variation in 2012 because 2012 is simulated for less than a month, whereas 2013 is simulated for 2 months. The variations in June 2012 and June 2013 are similar (about 5K in temperature for 2012 and 4K in 2013; about 11m/s in wind speed in 2012 and 7m/s in 2013).

Fig. C1. S1 cannot be seen in the OM1 figure. Are S1 and S5 overlapped? Scales in x-axes are different, i.e. PM10 July06-July31, PM1 June06-Aug05, OM1 Jun07-Aug05. It would be better if they were consistent, also with the meteorological figures (B1)

Yes, S1 and S5 are overlapped. Figure C1 is modified in the revised paper so that they are consistent with Fig B1.

Fig. C3: The modeled PM10 and PM1 $NO_3$ look exactly the same in all simulations, suggesting that coarse nitrate was not modeled. If this is the case, it should be mentioned in the methods section. Then there is no need to show the figure for PM10 $NO_3$ (same for $NH_4$ too).

The PM10 $NO_3$ and $NH_4$ are removed from Fig C3. Coarse nitrate is modelled, but the dust heterogeneous reactions to form nitrate and sulfate are not. The following sentence is added in the

method section "Dust heterogeneous reactions to form nitrate and sulfate are not taken into account".

Fig. D1: Please give the size fraction for the figures in the lower 2 panels.
The size fraction for the two lower panels of Fig D1 is the coarse fraction. It is added in the caption of Figure D1 in the revised paper.

Map for Cl+NO$_3$+NH$_4$ (please do not call it SIA) shows high concentrations (up to 24 microg/m$^3$). Please give some information which species is contributing more to this high level. Considering highly polluted Po Basin and high ammonia emissions in the region, high concentrations are very likely due to ammonium nitrate. It would make more sense to show maps of NO$_3$ and NH$_4$ separately and not to sum with Cl.
Maps of NO$_3$ and NH$_4$ are shown in Figure D5 of the revised paper. It shows that high concentrations of both inorganics are located over the same region stressing indeed the fact that they are due to the formation of ammonium nitrate.

Fig. D2. It would be useful to see also the absolute differences.
Maps representing the absolute differences are shown in Figure D6 of the revised paper.

Fig. D3: please separate Cl from NO$_3$ and NH$_4$. It is probably dominated by Cl.
Yes, it is dominated by Cl. A map of NO3 + NH4 is shown in Appendix D7.

Please revise the following references: Chrit et al 2017, Cholakian et al. 2017
These references are corrected in the revised paper.

**References added to the paper:**

EPA, U.: Guidance on the use of models and other analyses for demonstrating attainment of air quality goals for ozone, PM2.5, and regional haze, US Environmental Protection Agency, Office of Air Quality Planning and Standards, 2007.

Aksoyoglu, S., Ciarelli, G., El-Haddad, I, Baltensperger, U., and Prévôt, A. S. H.: Secondary inorganic aerosols in Europe: sources and the significant influence of biogenic VOC emissions, especially on ammonium nitrate, Atmos. Chem. Phys., 17, 7757-7773, https://doi.org/10.5194/acp-17-7757-2017, 2017.

A. Von Engeln, J. Teixeira A planetary boundary layer height climatology derived from ECMWF reanalysis data. J. Clim., 26 (2013), pp. 6575-6590, 10.1175/JCLI-D-12-00385.1

Seidel, D., Y. Zhang, A. Beljaars, J.-C. Golaz, A. Jacobson, and B.Medeiros, 2012: Climatology of the planetary boundary layer over the continental United States and Europe. J. Geophys. Res., 117, D17106, doi:10.1029/2012JD018143.

Aksoyoglu, S., Baltensperger, U., and Prévôt, A. S. H. (2016). Contribution of ship emissions to the concentration and deposition of air 5 pollutants in Europe. Atmos. Chem. Phys., 16:1895–1906.

Ntziachristos, L. and G. Papadimitriou, N. Ligterink, S. Hausberger, (2016). Implications of diesel emissions control failures to emission factors and road transport NOx evolution, Atmospheric Environment, 141, pp 542-551.g

---

## Author Comment (AC2) · 13 Jun 2018

The comment was uploaded in the form of a supplement:
https://www.atmos-chem-phys-discuss.net/acp-2017-915/acp-2017-915-AC2-supplement.pdf

---

## Author Comment (AC3) · 13 Jun 2018

This study presents the sensitivity of aerosols and their chemical composition over the Eastern Mediterranean as calculated by different model simulations performed in the framework of their ChArMEx experiment. The manuscript is very well organized and easy to follow, with a good level of English language. The manuscript is suitable for publication in ACP provided that the below minor comments are addressed in a revised version.

Material and Methods

1) Is the WRF model configured with 1-way or 2-way nested?

The WRF model was configured with 1-way nesting method.

This sentence is added in the revised version "…. To simulate WRF meteorological fields over the Mediterranean domain, 1-way nested WRF simulations are conducted on two nested domains …"

2) It would be good if more background is provided on how these different configurations for meteorology are designed.
The sentence:
"Before conducting the sensitivity study relative to meteorology (section 3) by using two different meteorological datasets, WRF is run with a number of different configurations, which are compared to measurements in section 3." is replaced by
"Before conducting the sensitivity study relative to meteorology (section 3) by using two different meteorological datasets, WRF is run with a number of different configurations, which are compared to measurements in section 3. In these configurations, the same physical parameterisations are used, but with different horizontal coordinates."
Furthermore, the description of the WRF configuration is added in the revised version:
"The WRF configuration used for this study consists of the Single Moment-5 class microphysics scheme (Hong et al., 2004), the RRTM radiation scheme (Mlawer et al, 1997), the Monin-Obukhov surface layer scheme (Janjic, 2003), and the NOAA Land Surface Model scheme for land surface physics (Chen et al., 2001). Sea surface temperature update, surface grid nudging (Liu et al., 2012, Bowden et al., 2012) options are activated. In the first configuration (WRF-Lon-Lat), horizontal …"

3) It is not clear from the text that both EMEP emissions and HTAP emissions are used in the simulations. Refer to Table 1.
The following sentence is added in the revised paper.
"… higher in HTAP emission inventory). HTAP emissions are used in the reference simulation and EMEP emissions for a sensitivity study as shown in Table 1.
Gaseous anthropogenic …"

4) Give more information on how the aerosols and organics are calculated in the model. Is it VBS that is used? Is the aerosol module operating on modal or sectional bins?

The aerosol modeling in this study is not based on the VBS approach but on a surrogate approach, and the aerosol size distribution uses a sectional approach.

A description of aerosol modeling is added in the revised paper after the description of the sensitivity studies on meteorology and sea-salt emissions: "…sodium and 4.22% of sulfate.

The SIze REsolved Aerosol Model (SIREAM; Debry et al., 2007) is used for simulating the dynamics of the aerosol size distribution by coagulation and condensation/evaporation. SIREAM uses a sectional approach and the aerosol distribution is described here using 20 sections of bound diameters: 0.01, 0.0141, 0.0199, 0.0281, 0.0398, 0.0562, 0.0794, 0.1121, 0.1585, 0.2512, 0.3981, 0.6310, 1.0, 1.2589, 1.5849, 1.9953, 2.5119, 3.5481, 5.0119, 7.0795 and 10.0 µm. The condensation/evaporation of inorganic aerosols is determined using the thermodynamic model ISORROPIA (Nenes et al., 1998) with a bulk equilibrium approach in order to compute the partitioning between the gaseous and particle phases of aerosols. Because the concentrations and the partitioning between gaseous and particle phases of inorganic aerosols (chloride, nitrate, ammonium) is strongly affected by condensation/evaporation and reactions with other pollutants, sensitivities of inorganic concentrations to hypothesis used in the modeling (thermodynamic equilibrium, mixed sea-salt and anthropogenic aerosols) are also performed (section 4.4.2)

For organic aerosols, the gas–particle partitioning of the surrogates is computed using SOAP assuming bulk equilibrium (Couvidat and Sartelet, 2015). The gas–particle partitioning of hydrophobic surrogates is modeled following Pankow (1994), with absorption by the organic phase (hydrophobic surrogates). The gas– particle partitioning of hydrophilic surrogates is computed using the Henry's law modified to extrapolate infinite dilution conditions to all conditions using an aqueous-phase partitioning coefficient, with absorption by the aqueous phase (hydrophilic organics, inorganics and water). Activity coefficients are computed with the thermodynamic model UNIFAC (UNIversal Functional group; Fredenslund et al., 1975). After condensation/evaporation, the moving diameter algorithm is used for mass redistribution among size bins. As detailed in Chrit et al. (2017), anthropogenic intermediate/semi-volatile organic compounds (I/S-VOC) emissions are emitted as three primary surrogates of different volatilities (characterized by their saturation concentrations $C*$: $\log(C*) = -0.04, 1.93, 3.5$). The ageing of each primary surrogate is represented through a single oxidation step, without NOx dependence, to produce a secondary surrogate of lower volatility ($\log(C*) = -2.4, -0.064, 1.5$ respectively) but higher molecular weight. Gaseous I/S-VOC emissions are missing from emission inventories, they are estimated here as detailed in Zhu et al. (2016) by multiplying the primary organic emissions (POA) by 1.5, and by assigning them to species of different volatilities. A sensitivity study where I-S/VOC emissions are not taken into account is also performed. "

5) Make it clear that dust emissions are not calculated in the model but only provided from the boundaries. Also provide information on how the boundary conditions are calculated. It is only in the results section that MOZART model is mentioned.

The boundary conditions are calculated from the simulation over Europe. The boundary conditions for the European simulation are calculated from the global model MOZART4.

Details about dust emission and boundary conditions are added after the paragraph describing the parameterization used for sea-salt emissions. "The boundary conditions for the European simulation are calculated from the global model MOZART4 (Horowitz et al., 2003) (https://www.acom.ucar.edu/ wrf-chem/mozart.shtml),, and those for the Mediterranean domain are obtained from the European simulation. Mineral dust emissions are not calculated in the model, but are provided from the boundaries."

6) Give more details on the SSE calculations. Is the surf zone included, how, or is only the open sea emissions calculated?

The sea-salt parameterizations used in chemistry transport models are built on open sea measurements. However, the same parameterization is used independently of the zone. This remark is added in the revised paper: "Sea-salt emissions are parameterized using Jaeglé et al. (2011) in the reference simulation and using the commonly-used Monahan et al. (1986) for a sensitivity study. These two parameterizations are based on open-sea measurements, but they are different in terms of the source function, "

7) Table 1 and 2 should be explained in the text and provide the motivation and the reasons for these different scenarios more clearly.

The different parameters, which are particularly attached to uncertainties, are detailed in the introduction. The section 2.1 aims to describe not only the set up but also alternative parameters/parameterizations commonly used in modelling. To clarify this point, the following sentence is added at the beginning of section 2.

"In order to simulate aerosol formation over the western Mediterranean, the Polair3d/Polyphemus air quality model is used, with the set-up described in Chrit et al. (2017) and summarized here. For parameters/parameterizations that are particularly attached to uncertainties (anthropogenic emissions,

meteorology, sea-salt emissions, modelling of condensation/evaporation), the alternative parameters/ parameterizations that are used in the sensitivity studies are also detailed. "
Section 2.1 is renamed "Simulation set-up and alternative parameterizations".

Results

8) Why the time series analyses for 2012 shown in the appendix? This would also show the extend of the dust break contribution to the levels.

All time series are shown in Appendix as specific pollution episodes are not studied. There was indeed a dust episode in 2012, but this episode is not discussed in details here, and a reference to the paper of Nabat et al. (2015) is added.

9) Explain Table 7 more clearly (in the caption maybe). What are the differences showing, period mean? What is the background of using normalized RMSE to show the sensitivity of the different inputs?
The periods used for the comparisons to measurements are now detailed in the caption of Table 7: "Comparisons of simulated $PM_{10}$, $PM_1$ and $OM_1$ daily concentrations to observations (concentrations and RMSE are in µg m$^{-3}$) during the summer campaign periods of 2012 (between 09 June and 03 July) and 2013 (between 07 June and 03 August)." We used a normalized RMSE rather than the RMSE to show the sensitivity of the different inputs in order to be able to compare the sensitivity of different pollutants which have different concentrations. For a pollutant with a relatively low concentration, the RMSE can be low but the sensitivity can be high compared to another pollutant with a higher concentrations. Normalizing the RMSE allows us to overcome this problem and to compare the sensitivity of the different pollutants.

10) As Table 7 shows, majority of the sensitivity simulation target 2013. Therefore, please also show the composition of PM in 2013 too to assist the discussions.
In Figure 3, the compositions of $PM_{10}$ and $PM_1$ for 2012 are replaced by the composition for 2013.

11) Add that these are observed composition in Figure 3 caption.

We added in the revised paper and in the text that Figure 3 shows the simulated compositions of $PM_{10}$ and $PM_1$.

" presented above. Figure 3 shows the simulated composition of …" and in the caption of Figure 3.

12) What does the 0.04±0.03 show in the Table 8?

This was a mistake. 0.04±0.03 is now removed from Table 8 in the revised version.

13) Page 15, Line 2. Normalized RMSE varies between 44 to 267%, not 48?
Yes, therefore 44% is replaced by 48 % in the revised paper.

Conclusions

What is the general conclusion of the study?

Particles are made of different compounds. The particle sources and the parameters influencing the concentrations are different depending on the compounds. These sources and parameters are identified here through 4 main conclusions for the different compounds/parameters.
The following sentence is added at the beginning of the conclusion section to clarify the aim of the study: "This work presents a sensitivity study to different input data and model parameterizations to better understand aerosol sources over the Mediterranean and the parameters influencing the aerosol concentrations. Aerosol sources are different depending on the aerosol chemical compounds. Comparisons to observations are performed at the ERSA station to estimate how realistic are the concentrations simulated with the different parameters. "
The conclusion was also rewritten to clarify the sources and the parameters influencing the concentrations and to better show the 4 main conclusion points:

- Sulfate originates mostly from maritime traffic. Furthermore, maritime traffic leads to the formation of oxidants that in turn enhance the formation of biogenic aerosols, with the potential formation of organic nitrate and organo sulfate.

- Organics are mostly from a biogenic origins. Even if the contribution of sea-salt emissions to organic concentrations is low, organic concentrations are strongly influenced by sea-salt emissions, because they partition between the gas and the particle phases and they are hydrophilic. This underlines the need to better characterize the properties (affinity with water) of secondary organic aerosols.

- Secondary pollutants, such as nitrate, ammonium and chloride, as the particle-phase concentrations are strongly influenced by the gas/particle phase partitioning, because a high percentage of their concentration is in the gas phase. This underlines the need to develop aerosol models able to represent accurately this gas-phase partitioning.

- There is a high sensitivity of secondary pollutants (inorganics and organics) to meteorology, stressing the importance of accurate meteorological modeling and the potential strong influence of climate change on the concentrations of these secondary pollutants.

.